# Retina-derived signals control pace of neurogenesis in visual brain areas but not circuit assembly

Shachar Sherman[1], Irene Arnold-Ammer[1], Martin W. Schneider [1],
Koichi Kawakami [2] & Herwig Baier [1]✉

Brain development is orchestrated by both innate and experience-dependent mechanisms, but their relative contributions are difficult to disentangle. Here we asked if and how central visual areas are altered in a vertebrate brain depleted of any and all signals from retinal ganglion cells throughout development. We transcriptionally profiled neurons in pretectum, thalamus and other retinorecipient areas of larval zebrafish and searched for changes in *lakritz* mutants that lack all retinal connections. Although individual genes are dysregulated, the complete set of 77 neuronal types develops in apparently normal proportions, at normal locations, and along normal differentiation trajectories. Strikingly, the cell-cycle exits of proliferating progenitors in these areas are delayed, and a greater fraction of early postmitotic precursors remain uncommitted or are diverted to a pre-glial fate. Optogenetic stimulation targeting groups of neurons normally involved in processing visual information evokes behaviors indistinguishable from wildtype. In conclusion, we show that signals emitted by retinal axons influence the pace of neurogenesis in visual brain areas, but do not detectably affect the specification or wiring of downstream neurons.

The assembly of neuronal circuits is classically considered an activity-dependent process[1]. Indeed, an abundance of evidence shows that central brain areas require input from the sensory surface for specification or maintenance of synaptic connections[2–7]. Retinal ganglion cells (RGCs), the sole output of the vertebrate retina, are a well-studied example of this principle. RGCs relay visual features to a range of anatomically well-defined central areas[8–11]. During development, they appear to also sculpt the assembly of downstream circuitry by imposing patterns of activity on their postsynaptic targets. Disrupting visual experience[12,13] or the waves of spontaneous activity that sweep across the retina[14] have been shown to lead to plastic changes in the mapping of connections in visual thalamus and cortex. In addition, even before they form synapses, RGC axons secrete morphogens, such as Sonic Hedgehog, which might regulate proliferation, commitment,

migration and differentiation of downstream neurons and glia[15,16]. Indeed, some of the changes in wiring observed after sensory deprivation may not be manifestations of synaptic plasticity, but rather the results of altered developmental signaling.

Modern transcriptomic techniques with single-cell resolution[17] offer the opportunity to explore systematically how the sensory surface influences gene expression, the production of neurons and the cell-type composition of the brain. Two recent single-cell RNA sequencing (scRNA-seq) studies tested if dark rearing of mice affected the development of visual cortex and found that apparently all cell types formed, with the exception of a class of interneurons in layer L2/3[18,19]. Notably, in these experimental paradigms, the contribution of spontaneous retinal waves to cortical development were not tested; nor could potential effects of RGC-derived secreted factors on

[1]Max Planck Institute for Biological Intelligence, Department Genes – Circuits – Behavior, Am Klopferspitz 18, 82152 Martinsried, Germany. [2]Laboratory of Molecular and Developmental Biology, National Institute of Genetics, and Department of Genetics, SOKENDAI (The Graduate University for Advanced Studies), Mishima, Shizuoka 411-8540, Japan. ✉e-mail: herwig.baier@bi.mpg.de

maturation of downstream circuitry be assessed, because RGC connections were left intact.

We asked if and how the sensory periphery influences the development of visual brain areas in zebrafish, an organism whose brain is continually growing into adulthood and undergoes plastic changes in response to learning, sensory deprivation and injury[20–24]. The eyes of zebrafish are open at all stages, and visually evoked responses are seen as early as 2.5 days after fertilization (dpf)[25], affording sensory-evoked and spontaneous activity ample opportunity to shape downstream circuitry. We transcriptionally profiled cells in the thalamus and pretectum, two divisions of the diencephalon that receive retinal input[26,27], as well as in the tectum and some of its associated nuclei in midbrain and hindbrain. Marker genes were anatomically localized by fluorescent in situ hybridization chain reaction (HCR-FISH) labeling. We then used the *lakritz* mutant, which carries a null allele of the RGC fate-determinant factor Atoh7 (Ath5) and consequently lacks all RGCs[28]. We found that removal of retinal-axon derived cues throughout development delays the cell-cycle exit of progenitors, but, perhaps surprisingly, leaves the differentiation, anatomical location and wiring of neuronal cell types largely unaffected. Thus, depletion of any kind of retinal input did not detectably alter composition or configuration of neuronal circuitry in retinal target areas.

## Results

### scRNA sequencing and multiplexed HCR-FISH labeling generate a molecular brain atlas

The *HGn12C:GFP* reporter line (short for *Et(HGn12C:GFP)*) was originally isolated in an enhancer-trap screen[29] and labels most, or all, neurons in pretectum, dorsal thalamus and ventral thalamus at 6 dpf, as well as several anterior nuclei of hypothalamus and subsets of neurons in habenula, tectum, nucleus isthmi and medulla (Fig. 1a). Non-neuronal cells labeled in *HGn12C:GFP* include oligodendrocytes and neuronal progenitor cells. We transcriptionally profiled 123,224 fluorescently sorted cells from 6 dpf larvae, of which 95,122 passed quality control. Glutamatergic and GABAergic neurons were clustered separately (Fig. 1b–f). This resulted in 40 glutamatergic and 37 GABAergic clusters, each identifiable by one, or a combination of few, specific marker genes (Fig. 1g, h).

Subpopulation-specific expression of markers was verified using a multiplexed wholemount fluorescent in situ hybridization protocol (HCR-FISH; Fig. 2, Supplementary Fig. 1). Labeling patterns were registered to a standard reference brain (Fig. 2, Supplementary Fig. 1) and related to classically annotated brain regions in the Max Planck Zebrafish Brain Atlas[30,31] at mapzebrain.org (Table S1).

### Gene expression domains resolve cell types in thalamus, pretectum and tectum

Our dataset provides a rich resource for explorations of the zebrafish visual brain. In the thalamus, individual cell types were identifiable by the expression of *crhb*, *crhbp*, *cckb*, *atf5b*, *npy*, *cort/sst7* and *sst1.2* (Fig. 2a, c, Supplementary Fig. 1a, c). Interestingly, the neuropeptide-encoding gene *pth2* was reported to be expressed in a small subset of thalamic cells involved in mechanosensation[32]. Our data uncovered that there are two thalamic *pth2*+ populations, one also expressing the neuropeptide Cortistatin (encoded by *cort/sst7*), in addition to a distinct population in the tectum (Fig. 2a). We also found that the genes *pax7a*, *aldh1a2*, and *cabp5b* are relatively specific markers for subsets of pretectal cells (Supplementary Fig. 1c, d). Expression of the transcription factor Sp9 offers a molecular landmark separating the dorsal thalamus from the pretectum (Supplementary Fig. 1c). The genes *calb2a*, *nfixb*, *esrrb*, *sox14* and *zic2a* each label non-overlapping M1 cell populations (Supplementary Fig. 1b; Table S1). The pretectal migrated area M1 is partially homologous to the mammalian accessory optic system[33]. The gene *pax6a* additionally shows expression in two nuclei bordering on M1 (mapzebrain.org; Table S1). In the midbrain tectum,

the gene *pou4f2* encodes a transcription factor specifically labeling glutamatergic neurons, while cells residing in the tectal neuropil express *gjd2b*, which encodes a gap-junction protein (Fig. 2f). Lastly, we also identified markers for other visual centers in the midbrain, such as *BX088*, an unknown gene, and *sema3fb*, which are both expressed specifically in the nucleus isthmi[34] (Table S1). These discoveries serve as examples illustrating the resolving power of our approach. In conclusion, the molecular cell-type catalog and spatial gene expression atlas presented here reveal the complex genetic architecture of the diencephalon and mesencephalon in larval zebrafish.

### Neuronal cell types are unaltered in thalamus, pretectum and tectum of *lakritz* mutants

The *lakritz* mutation disrupts the gene encoding the basic helix-loop-helix transcription factor Atoh7 (Ath5), which is critical for RGC cell fate determination and is not expressed outside of the retina[28]. Homozygous mutants are viable and show normal behavior, except for their blindness. The *lakritz* mutation is completely recessive[28]. We crossed *HGn12C:GFP* into a *lakritz* mutant background and sequenced 20,221 fluorescently sorted mutant cells and 25,687 WT sibling cells, with 17,029 and 18,443 cells passing quality control, respectively. Strikingly, we could not detect a major difference in the transcriptomic profiles between the different genotypes (Fig. 3a, b): every cell cluster that developed in wildtype (WT) was also present in mutants and phenotypically normal siblings. Comparing individual replicates of each genotype excluded a possible confound of batch effects (Supplementary Figs. 2 and 3). The similarity of *lakritz* to WT clusters was also robust after bioinformatic separation of glutamatergic and GABAergic neurons (Supplementary Figs. 4–7), with few exceptions (see "Methods"). Embedding all samples together in a single UMAP plot did not reveal a difference between samples (Fig. 3b, c). We also color-coded *lakritz* cells based on results from *lakritz*-only pre-clustering and were able to see the same clusters form in an embedding containing all groups. This visualization excludes the possibility that a larger WT sample would force the *lakritz* population to embed similarly (Supplementary Fig. 8).

To explore whether the UMAP dimensionality reduction was sensitive to subtle differences, we designed a computational pipeline simulating a cell-type ablation experiment. For this, an algorithm omitted *lakritz* cells belonging to a single glutamatergic or GABAergic cluster from the original count matrix. Then, the artificially truncated matrix was reclustered and visualized in a blinded fashion. This "in-silico cluster ablation" approach confirmed that visual inspection of UMAP plots would have detected missing clusters with high confidence (Supplementary Figs. 9 and 10; see "Methods" for details).

Next we asked if we could detect differences in transcript abundance between genotypes. A large number of genes indeed varied between *lakritz* and WT. These differences were slightly less pronounced between the two WT groups. However, only a few of the potentially dysregulated genes pass a threshold defining cell-type markers, and the top principal components (PCs) are identical across groups (Supplementary Fig. 11; see "Methods" for details). Moreover, we could not detect a global, or cluster-specific, drift in gene expression related to the *lakritz* genotype, suggesting most, if not all, pretectal and thalamic cell types are present and largely unaltered despite absence of RGCs.

### Cell-type proportions are largely unchanged, but some marker genes are locally downregulated in *lakritz* brains

Next, we tested whether we could identify differences in the relative proportion of specific cell-type clusters. We could not detect significant differences in the abundance of specific clusters between samples (Fig. 4). To test whether some cell types show more severe transcriptome alterations than others by the absence of RGCs, we

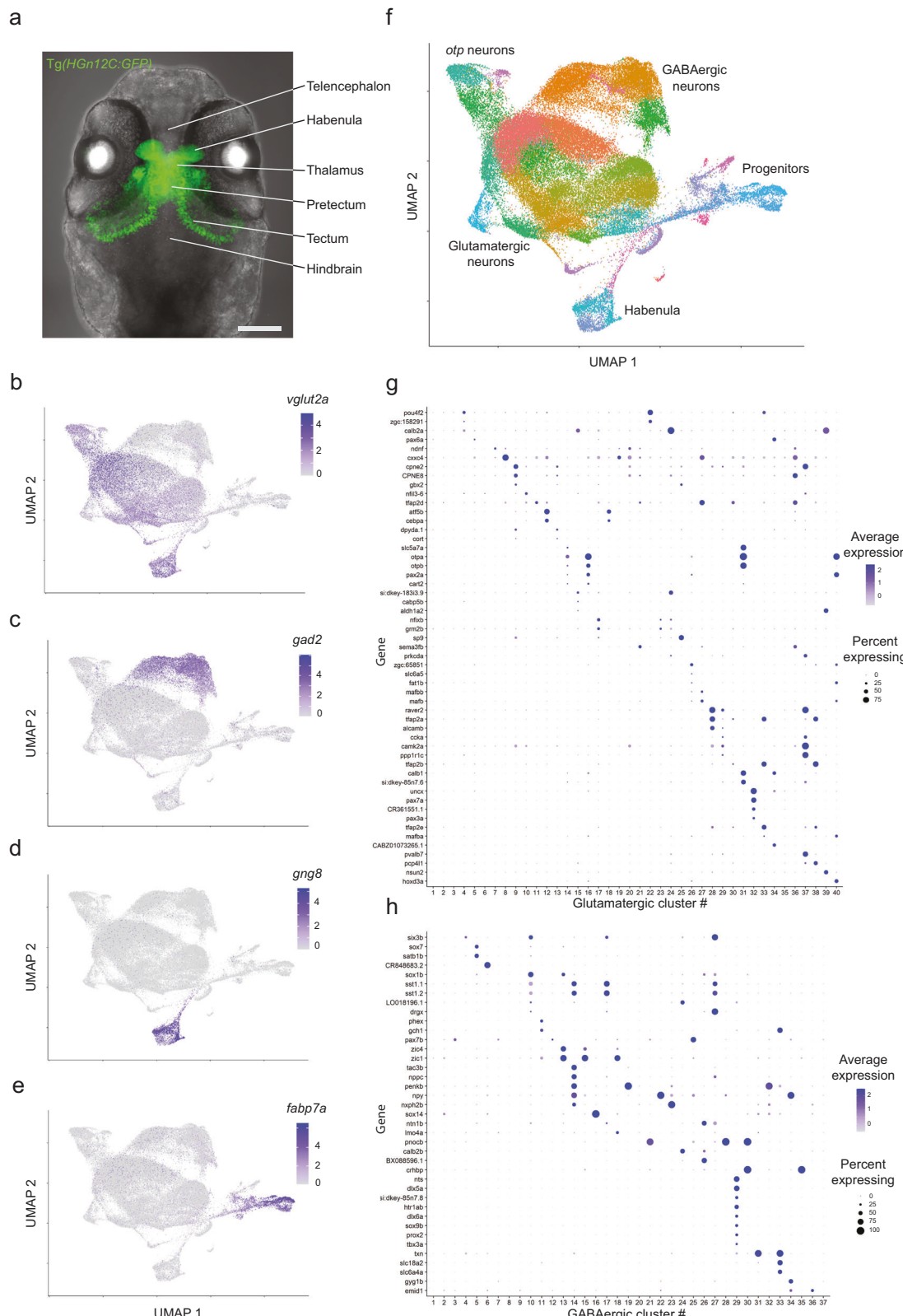

**Fig. 1 | Transcriptional profiling reveals cell types across central visual areas.**
**a** Maximum z-projection of the *HGn12C:GFP* expression pattern in green. Transmitted light is in gray. Lines and labels annotate major anatomical brain areas of the larval brain (scale bar = 100 μm). **b**–**e** Gene expression plots of cells embedded in UMAP space (*vglut2a*, glutamatergic neurons; *gad2*, GABAergic neurons; *gng8*, habenula neurons; *fabp7a*, progenitors). (f) UMAP embedding of all sequenced cells. Color-coding represents different clusters. Text labels adjacent cell classes. UMAP space is the same as in (**b**–**e**). **g**, **h** Markers for glutamatergic (top) and GABAergic (bottom) clusters. Color shade represents the average level of marker expression in a cluster (average expression). Dot size represents the percentage of cells expressing the marker in a cluster (percent expressing).

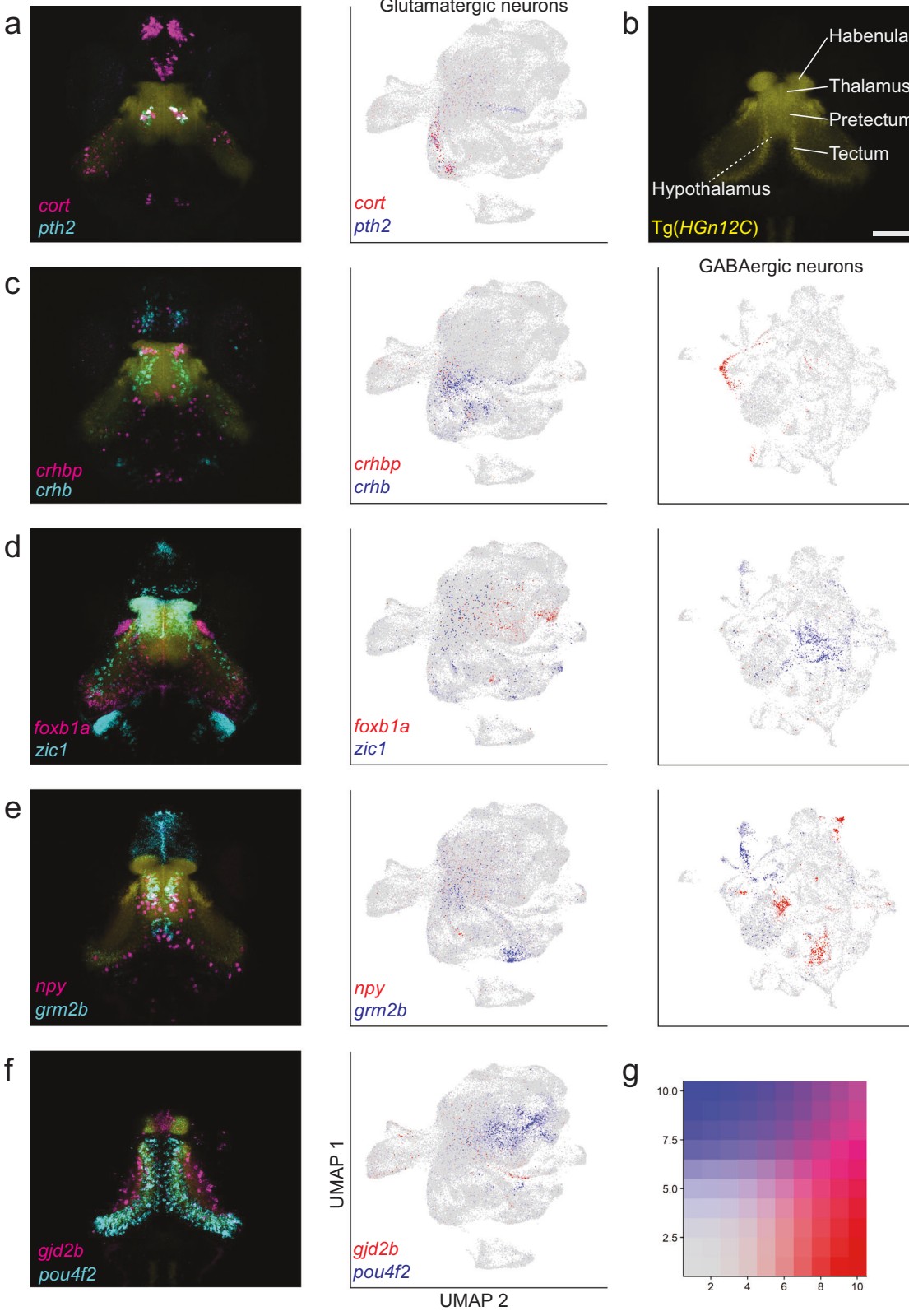

**Fig. 2 | HCR-FISH labeling uncovers the molecular architecture of central visual areas.** Marker genes label cells in distinct anatomical regions. **a**, **c**–**f** Substack maximum z-projections of registered HCR-FISH stains. From top to bottom (left): **a**, **c** Selected thalamic markers: *cort*, *pth2*, *crhbp*, *crhb*. **d**, **e** Selected pretectal markers: *foxb1a*, *zic1*, *npy*, *grm2b*. **f** Selected tectal markers: *gjd2b*, *pou4f2*. Alongside each stain are UMAPs (same as in Fig. 1), showing the clustering of glutamatergic (middle) and GABAergic cells (right) that express the marker. UMAPs with no clustered expression were omitted. **b** In yellow, *HGn12C:GFP* background stain used for registration (scale bar = 50 μm; applies to all images). For all stains, at least three larvae were imaged. Additional stains and anatomical annotations are available in the zebrafish brain atlas[31] at mapzebrain.org. **g** Gene expression look-up matrix for combined plots.

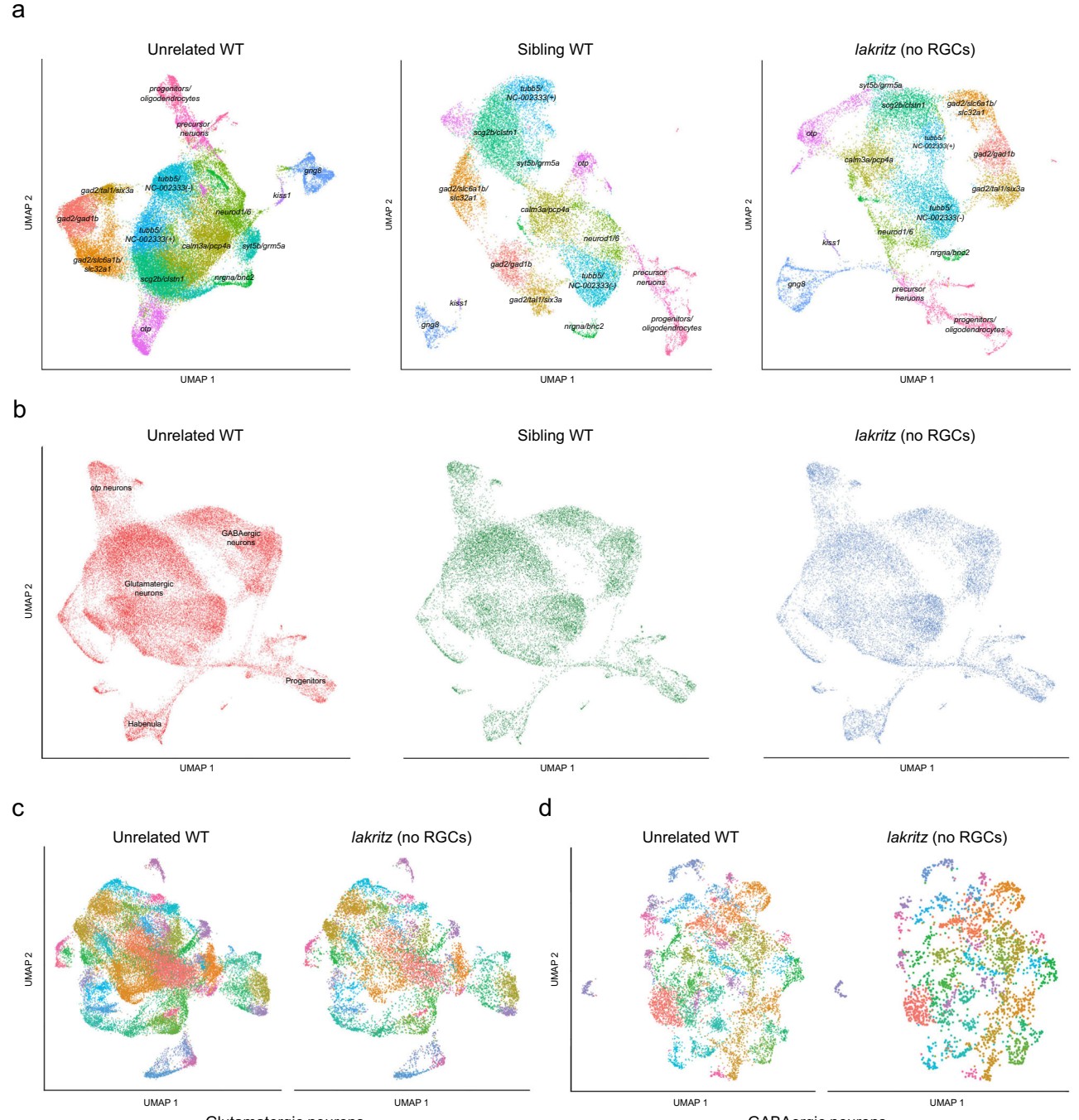

**Fig. 3 | Cell-type diversity of central visual areas emerges in the absence of retinal input. a** UMAP embedding of different genotypes clustered independently (unrelated WT, left; sibling WT, middle; *lakritz*, right), color-coded by cluster identity. Clusters are labeled with text. GABAergic clusters express *gad2*. The remaining clusters, excluding progenitors and precursor neurons, are glutamatergic. **b** UMAP embedding of all cells color-coded by genetic background. Text labels adjacent cell classes. Clustering of glutamatergic (**c**) and GABAergic cells (**d**). Presented side-by-side are WT and *lakritz* cells of the same clusters. For quantitative analysis, see supplement.

applied an analysis used before to uncover such clusters[35] (see "Methods"). In short, for each cluster we compared the change in expression of marker genes between WT and *lakritz* cells compared to the change in expression for all the genes detected in a given cluster. We used p-values calculated in this analysis to color-code our UMAP, highlighting cell types showing the most pronounced transcriptional changes (Supplementary Fig. 12). We then compared the expression patterns of the top cluster-specific markers in WT and *lakritz* by HCR-FISH[36]. This analysis revealed complex region- and gene-specific

differences in expression (Supplementary Fig. 13). For example, cells of the dorsolateral central pretectum strongly express *cabp5b*; in *lakritz* mutants, this expression is reduced. On the other hand, *aldh1a2* is unchanged in the same population, but diminished in a population of more medial pretectal cells (Supplementary Fig. 13b, c). The *calb2b* gene is downregulated in *lakritz* mutants across the tectum (Supplementary Fig. 13d). The *crhbp* marker is expressed in multiple nuclei across the forebrain, but noticeably downregulated in a single thalamic nucleus in the mutant (Supplementary Fig. 13e).

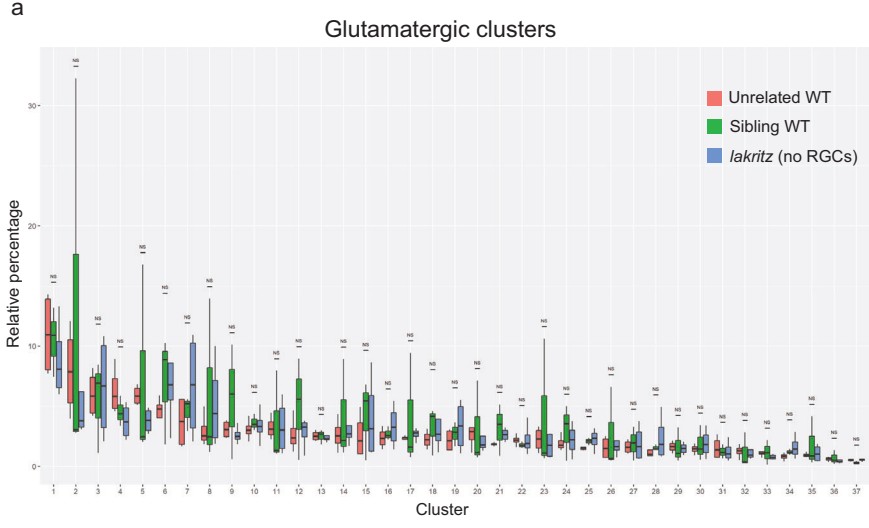

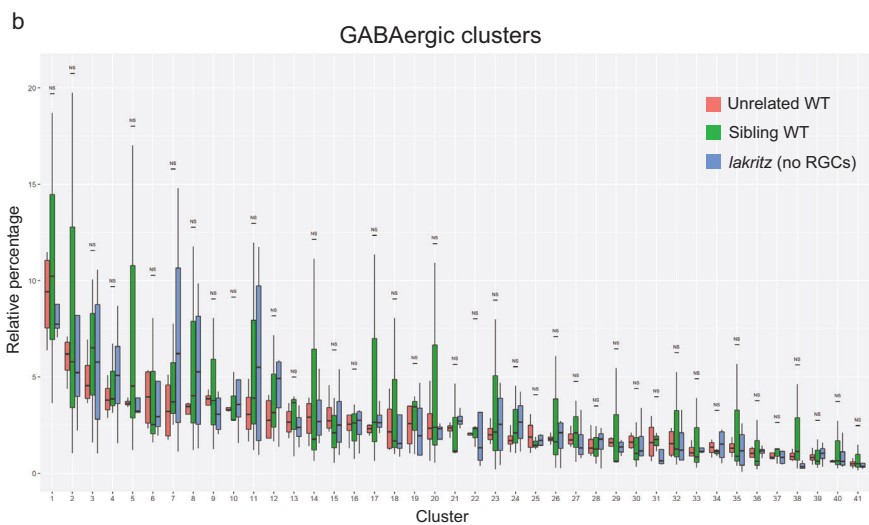

**Fig. 4 | Relative cell-type proportions are largely unchanged in absence of RGCs.** Bar plots showing for each cluster the variance in relative percentages across replicates (**a** glutamatergic clusters; **b** GABAergic clusters). For each cluster, the variance is shown for all genotypes (red, unrelated WT; green, sibling WT; blue, *lakritz*). The p-values were calculated using a two-sided Wilcoxon signed-rank test and corrected for multiple testing using the Bonferroni correction (unrelated WT: $n = 4$ independent experiments; sibling WT: $n = 3$ independent experiments; *lakritz*: $n = 4$ independent experiments).

## Differentiation trajectories are unaffected, but progenitor commitment is delayed, in *lakritz* brains

In addition to neurons, the *HGn12C:GFP* reporter also labels (mitotic) progenitors and (postmitotic, differentiating) precursors. These groups can be clearly distinguished based on their transcriptional profiles (Fig. 1e, f). In our UMAP embedding, progenitors and differentiated neurons are connected by a single "thread-like" cluster of differentiating precursors, reflecting the gradual transition of the single-cell transcriptomes from uncommitted, mitotic to postmitotic, fate-committed terminal stage. Further visualization revealed a split of late-stage precursors into a glutamatergic (expressing *neurog1*) and a GABAergic (expressing *ascl1b* and *sox2*) branch in all samples (Fig. 5A, Supplementary Fig. 14a, b). In addition, a subset of the glutamatergic precursors branch off to a habenula fate (expressing *cxcr4b*; Supplementary Fig. 14c). Confirming a previous report[37], mature habenula neurons fall into one of two subpopulations: *gng8*-positive or *kiss1*-positive cells (Fig. 5A, Supplementary Fig. 14c). Subpopulations of precursor neurons express two kinds of marker genes: sustained "cell-type markers", which are characteristic of differentiated neuronal clusters, probably indicating commitment to a specific fate, and

"transient markers", which are downregulated in mature neurons and may contribute to the developmental transition (Table S2).

This resolution afforded us the opportunity to investigate if the absence of RGCs altered developmental trajectories. In *lakritz* mutants, differentiation pathways are akin to WT (Fig. 5B, C). Strikingly, however, progenitors and early precursors are enriched relative to neurons in *lakritz* mutants (WT: 82.6% neurons, 4.4% progenitors and early precursors, ratio = 19; *lakritz*: 73.5% neurons, 7.9% progenitors and early precursors, ratio = 9; Chi-squared test: *p* value < 2.2 × 10^{-16}).

## Trajectory inference analysis reveals a glial bias and a slower cell cycle in *lakritz* progenitors

We reclustered a dataset that included only progenitors and precursors and applied a set of computational methods for trajectory inference[38–40] (Fig. 5C–E, Supplementary Fig. 15). RNA velocity largely recapitulated progenitor trajectories into either glutamatergic or GABAergic precursors, but additionally underscored a third trajectory into glial precursors (Fig. 5C, Supplementary Fig. 15a–d). Latent-time analysis shows initial and terminal macrostates and allows comparing

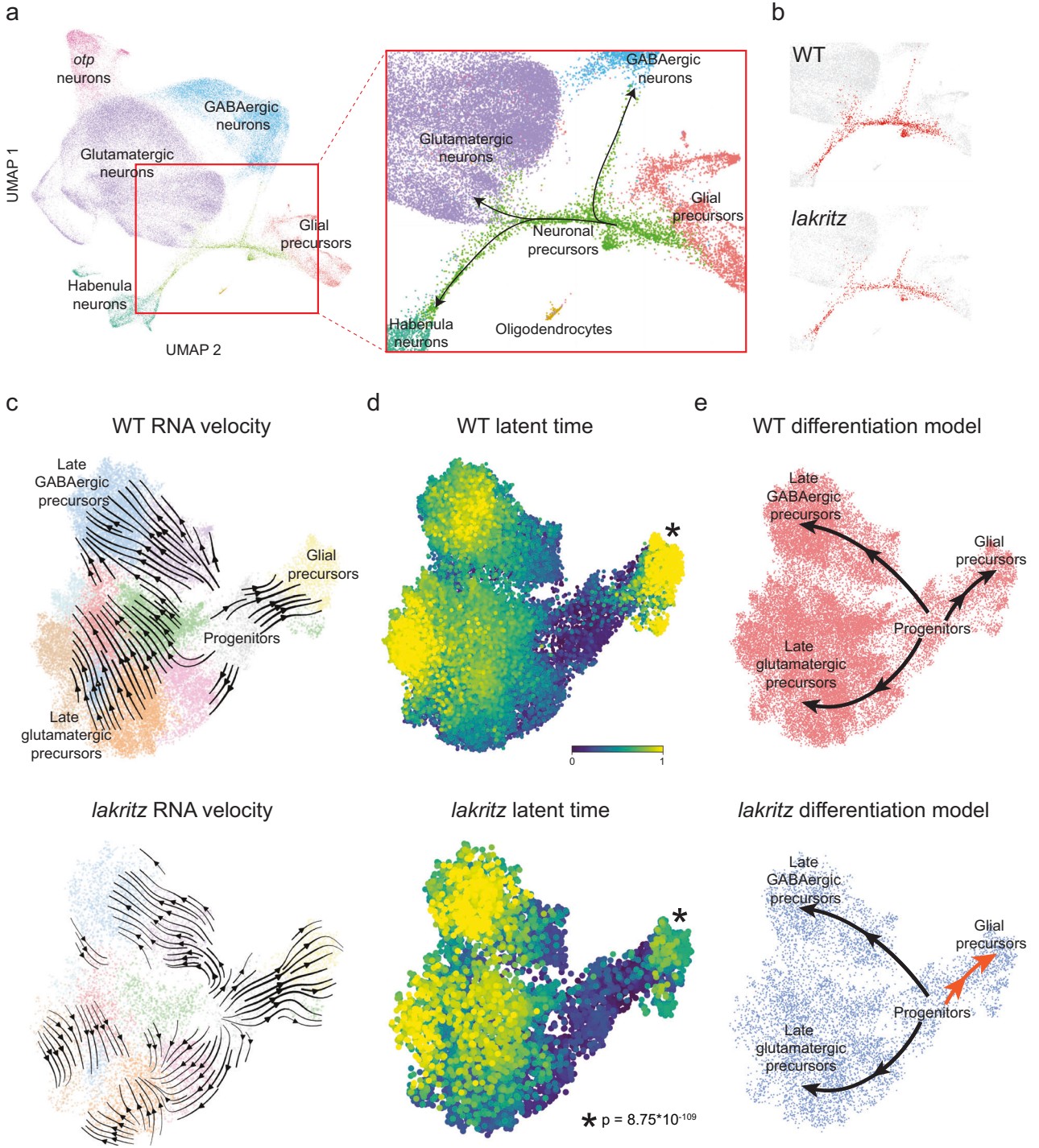

**Fig. 5 | Differentiation trajectories through transcriptomic space are conserved despite absence of RGCs but differ in speed. A** Left, UMAP embedding of all cells, color-coded by cell identity. Text labels adjacent cell classes. Right, enlarged area (red square, left) showing a class of neuronal precursor cells differentiating into major neuronal classes. **B** Same enlarged area as in (a, right), but highlighting in red neuronal precursors from either WT (top) or *lakritz* (bottom). Precursors from both groups show similar differentiation paths. **C**–**E** Comparison of WT (top) and *lakritz* (bottom) by trajectory inference analysis. **C** RNA velocity analysis including mitotic progenitors and postmitotic precursors. Clusters are the same across groups. **D** Latent time analysis informed by RNA velocity. Initial and terminal states were calculated independently across groups. *P*-value calculated by comparing latent time for the glial precursor terminal state (asterisk adjacent) between groups using the Wilcoxon signed-rank test. **E** Simplified models for differentiating progenitors and precursors across groups. In *lakritz* mutants, precursors show a faster transition (velocity) from progenitors to glial precursors (dark orange arrow).

the relative time it takes for a putative "root cell" (progenitor) to differentiate towards individual terminal states[39] (glutamatergic, GABAergic, or glial precursors). This analysis uncovered that without RGC-derived cues, progenitors take a much more rapid transition into glial differentiation, but not neuronal ones (Fig. 5D, E). Velocity length

analysis, which more directly measures expression transition speeds, showed similar results while emphasizing a slight slow-down in neuronal production (Supplementary Fig. 15e, f).

In our reduced dataset, the faster transition into glial precursors becomes detectable immediately after the final cell cycle exit. We

reasoned that this could potentially result from earlier effects present in mitotic progenitors. To explore this possibility, we generated a second, smaller dataset including only progenitors (Fig. 6a). Pseudo-time analysis revealed a temporal cyclical progression structure embedded in our data with a single-cell-cycle exit point (Fig. 6b–d). RNA velocity confirmed that these trajectories are similar between WT and *lakritz* mutants (Fig. 6e, f). Visualizing velocity lengths revealed that WT progenitors show a relatively homogenous speed with a significant slow-down towards cell-cycle exit and G1 arrest (Fig. 6g). In contrast, *lakritz* progenitors show a slower cell cycle. At the exit point, *lakritz* progenitors appear to accelerate towards G1 arrest over cell-cycle exit (Fig. 6f, i). This suggests that compared to WT, *lakritz* progenitors have a longer cell cycle, show a lower preference to exit the cell cycle, and a higher preference towards G1 arrest (Fig. 6j, k). Quantifying the proportion of progenitors in each of the cell-cycle phases supports this model: Compared to WT, *lakritz* progenitors show a significant increase in progenitors in G1 and a decrease in progenitors in either S or G2/M phases (Fig. 6l).

### Expression of progenitor and early precursor markers is altered in *lakritz* mutant brains

Next, we used HCR-FISH to investigate if altered progenitor behavior is detectable at the level of marker gene expression. We chose marker genes expressed in progenitors and early precursors (*cyp19a1b, fabp7a, gfap, her4.1, p27, pcna* and *s100b*) and differentiating precursor neurons (*ascl1b, neurod1, neurog1* and *sox2*) across 11 proliferative and adjacent regions (Table S3). The most significant differences were observed in areas that normally receive retinal input. In *lakritz* mutants, the glial precursor marker *fabp7a* is downregulated across the diencephalon and, together with the glial maturation marker *s100b*, in tectal ventricles (Fig. 7a, d; Table S3), suggesting that glial differentiation is altered. Likewise, the proliferation marker *pcna* shows significant downregulation in *lakritz* in the thalamus (Table S3), consistent with overall fewer cells in S phase.

To begin to explore the developmental stage when neuronal differentiation could be most affected by absence of retina-derived cues, we also tracked the expression of a glutamatergic (*neurog1*) and a GABAergic marker (*sox2*) from 3 to 7 dpf (Table S4). We found that these markers are differentially dysregulated in *lakritz* at 6 and 7 dpf in the tectum, pretectum and thalamus (Table S4). Together, these findings point to a subtle, but lasting dysregulation of neurogenesis in areas of the brain lacking retinal input.

The regulation of neurogenesis involves the intersection of many factors, both spatially and temporally[15]. Sonic hedgehog (Shh) expression in the retina and binding to the Patched receptor, for example, is required for RGC development[41,42]. At a later stage, Shh is produced and secreted by RGCs and sensed by retinal precursor cells (RPCs) to ensure a correct ratio of RGCs to RPCs. Shh is also expressed in the prechordal plate, which extends into ventral midbrain and forebrain, and has a prominent expression domain in the zona limitans intrathalamica, a band that separates the two thalamic prosomeres. Similarly, Wnt factors, expressed by RGCs, are involved in neurogenesis by regulating the proliferation of stem cells in the retinal margins[43] and might also influence differentiation of downstream circuitry. We performed stains for *shha*, and *shhb*, as well as genes encoding Shh receptors (*ptch1* and *ptch2*), and components of the Wnt signaling pathway (*wnt3, wnt3a, axin2* and *lef1*). Strikingly, genes that encode Shh, Ptch and Wnt were reduced in several areas lacking retinal input (Fig. 7b, c, e, f; Table S3). These effects were region-specific and might account for some of the differences in progenitor and precursor behavior in *lakritz* mutants.

### Functional circuitry in the pretectum forms in the absence of retinal input and supports behavior

We set out to test whether visual networks form properly during development without the input from RGCs. We used optogenetics to probe functional connections underlying two visually guided behaviors: prey capture and the optokinetic response (OKR). Detection of prey is mediated by neurons extending dendrites into the retinal projection field AF7[44,45]. These receive synaptic input from RGCs and are upstream to the premotor and motor circuitry involved in initiating pursuit and capture of prey (Fig. 8a). We used the *pvalb6* enhancer-trap line, which labels neurons around AF7[44], to drive expression of channelrhodopsin (CoChR2) in WT and *atoh7* morphants, which lack mature RGCs and thus phenocopies the *lakritz* mutant (Fig. 8a). Photostimulation of AF7 neurons in the *pvalb6* line triggers so-called J-turns, which are tail movements that orient larvae towards prey[46] (Fig. 8b). In total, we were able to elicit J-turns in a similar proportion of WT and *atoh7* morphants (5 of 7 and 4 of 7, respectively). The onset of J-turns after photostimulation varied more across *atoh7* morphants compared to WT (Supplementary Fig. 16). This could be due to a developmental defect or to differences in activation thresholds, or some other physiological change in RGC-depleted brains.

Next, we sought to test whether the circuitry underlying OKR is established without retinal input. Previous work had identified the direction-selective circuitry in the ventro-anterior pretectum as a hub driving the OKR in zebrafish larvae[33,47,48]. Photostimulation of channelrhodopsin (ChR2)-expressing neurons in the enhancer-trap line *Gal4s1026t*[49] via an optic fiber elicited a sequence of slow pursuit eye movements and saccades typical of the OKR[48]. We crossed *Gal4s1026t* and *UAS:ChR2-mCherry* into a *lakritz* mutant background and exposed fish larvae sequentially to moving gratings in a visual arena and to targeted photostimulation (Fig. 8c,d). As expected, WT larvae, but not *lakritz* mutants, respond to visual stimulation (Fig. 8e). Optogenetic activation of the pretectal area, on the other hand, evokes full OKR-like behavior in both WT and mutants (Fig. 8e, Supplementary Fig. 17).

To verify that the direction-selective cells driving expression of ChR2 in *Gal4s1026t* overlap with cells labeled in the *HGn12C:GFP* reporter line, we sequenced them and introduced their transcriptomes into our cell-type catalog (Supplementary Fig. 18a). We also co-expressed the two reporters in a triple-transgenic fish (Supplementary Fig. 18b) and co-registered them within the Max Planck Zebrafish Brain Atlas (mapzebrain.org; Supplementary Fig. 18c). Together, these analyses showed that the cells expressing neuronal markers in the *Gal4s1026t* domain were a subset of those present in the *HGn12C:GFP* forebrain catalog. While *Gal4s1026t* labels additional non-neuronal cells (likely differentiated glia; Supplementary Fig. 18a), its expression is restricted to a smaller pretectal subvolume than is labeled by *HGn12C:GFP*.

## Discussion

The zebrafish central nervous system develops rapidly, grows through adulthood and shows substantial plasticity[20]. The role that sensory inputs play in shaping the assembly of zebrafish neuronal circuitry has not been explored systematically. Here we studied the composition and function of visual forebrain centers in *lakritz* (*atoh7, ath5*) mutants, employing single-cell transcriptional profiling and optogenetic reconstitution experiments. Extensive work over the past three decades has firmly established that the cell-autonomous deficit in the *lakritz* mutant is restricted to the retina: RGCs, the first-born neurons in the retina, fail to develop; instead, amacrine cells are produced in greater numbers[28]. As a result, the central brain in these mutants is entirely cut off from retina-derived signals throughout development, be it molecular factors secreted by RGC axons, their growth cones or presynaptic terminals, waves of spontaneous neural activity, or visual experience. This genetic lesion is thus more encompassing than conventional disruption of visual inputs, such as enucleation (removal of eyes), sensory deprivation (by dark rearing), synchronization of activity (by stroboscopic light), or pharmacological perturbations.

We first resolved the diversity of cell types in zebrafish forebrain and midbrain areas that receive RGC projections. Single-cell

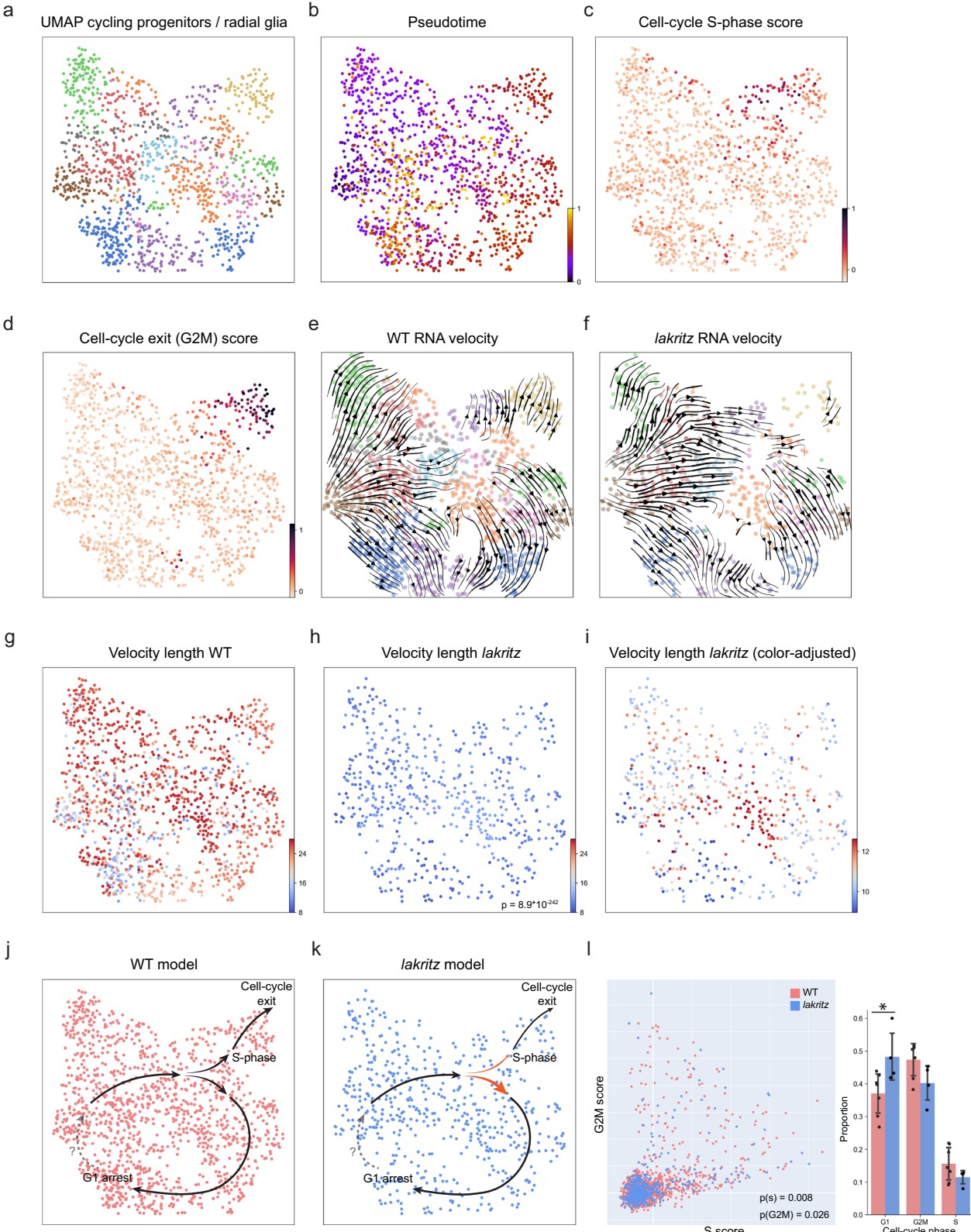

transcriptional profiling was carried out on cells labeled in the reporter line *HGn12C:GFP*[29] which, as shown here, labels both neurons and neural progenitors in the pretectum, the ventral and dorsal thalamus (prethalamus and thalamus), as well as scattered cells in anterior hypothalamus, tegmentum, medulla, nucleus isthmi, and tectum. Clustering of single-cell transcriptomes at 6 dpf revealed 77 putative cell types, as well as a pool of (dividing) progenitors and three groups of (postmitotic) precursors – glutamatergic, GABAergic, and glial.

Their stages of commitment and differentiation could be traced in the dataset by computational inference of trajectories through transcriptional space and RNA velocity analysis. This analysis uncovered marker genes, which were differentially expressed among clusters and in reproducible patterns. Among a wealth of interesting observations, our spatial transcriptomics analysis revealed that the larval zebrafish thalamus is organized in anatomical clusters likely corresponding to single brain nuclei. This is in contrast to a similar analysis of the adult

**Fig. 6 | Mitotic progenitors progress more slowly through the cell cycle and exit it at a reduced rate in absence of retinal input. a–l** Trajectory inference analysis performed on a cluster of mitotic progenitors. **a** UMAP embedding of cells color-coded by in-subset cluster identity. All subsequent panels use the same UMAP embedding. **b** Pseudotime analysis shows temporal progression (transition) between neighboring clusters. **c, d** Cells are color-coded according to their cell-cycle score. **c** Score of cell-cycle S-phase genes. **d** Score of cell-cycle G2M-phase genes. G2M high-scoring cells show a putative transition point from mitotic to postmitotic progenitors. RNA velocity analysis performed on WT (**e**) and *lakritz* (**f**) cells independently. RNA velocity show a similar temporal progression across groups. Velocity length inferred from RNA velocity shown for WT (**g**) and *lakritz* (**h, i**) cells. Color scale is the same for both groups in (**g, h**). **i** Color-scale adjusted to 5th and 95th percentile of values in the *lakritz* dataset. Lower velocity length shows that transition speed in *lakritz* is reduced overall in mitotic progenitors. In (**i**)

higher-scoring cells show a putative bias towards G1 arrest over cell-cycle exit in the *lakritz* dataset. *P*-value calculated by comparing velocity length between WT and *lakritz* using the Wilcoxon signed-rank test. **j, k** Simplified models for mitotic progenitors across groups. In *lakritz*, mitotic progenitors show a bias towards G1 arrest over cell-cycle exit (dark orange arrow). Dashed arrow shows a putative transition from G1 arrest to active cycling. **l** A scatter plot showing the distribution of cell-cycle scores for each cell in the WT (red) and *lakritz* (blue) datasets. *P* values were calculated by comparing scores distribution between groups for each axis using the Wilcoxon signed-rank test. In *lakritz*, a significantly larger proportion of cells show low and negative scores on both axes, suggesting an increase in G1 cells. (Right) Bar plot summarizing the cell-cycle assignment for each cell shows a significant increase in the proportion of G1 cells in the *lakritz* dataset ($p = 0.0376$, two-sided Wilcoxon signed-rank test; $n = 1333$ cells, collected over 11 independent experiments. Data bars are mean ± SD).

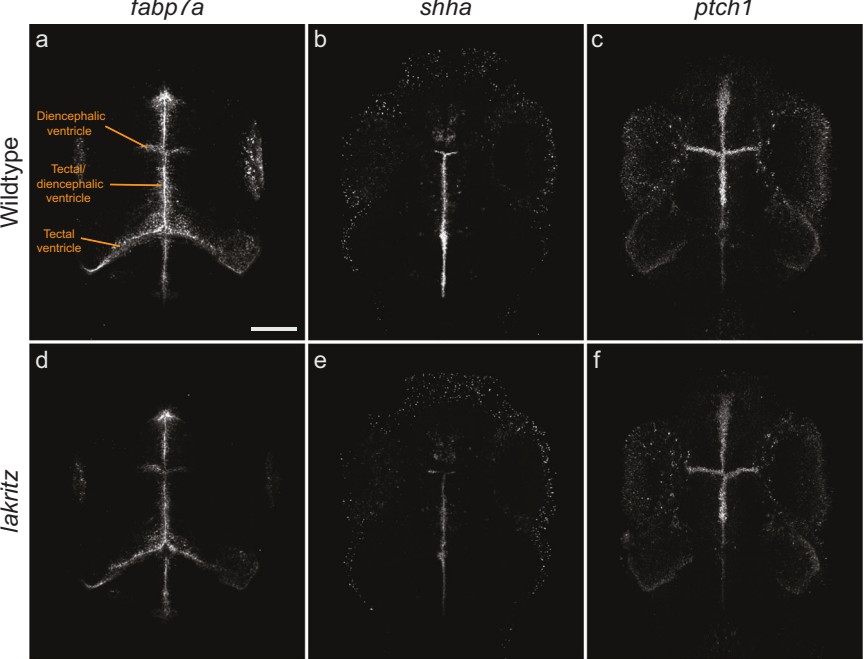

**Fig. 7 | Some developmental markers are dysregulated in the ventricular zones of *lakritz* mutants.** The zones lining the ventricles (indicated) in the midbrain and forebrain contain proliferative and differentiating cell populations. Overall patterns are unchanged in *lakritz* mutants (**c, d, f**) compared to wildtype (WT; **a, b, c**), although expression levels are altered. See also Table S3. The marker of terminally differentiated glia, *fabp7*, is downregulated in *lakritz* mutants (**a, d**). The genes

encoding the mitogen Shha (**b, e**) and its receptor Patched1 (**c, f**) are reduced in *lakritz* mutants. Maximum projections of HCR-FISH labeling patterns from the same co-registered substacks are shown to allow direct comparisons of local expression levels across fish. Scale bar = 100 μm. All experiments were repeated three times with similar results.

mouse thalamus, which has been reported to lack gene expression domains that demarcate subdivisions[50], but is in agreement with the neuroanatomical literature that has long recognized separable nuclei. The positions of genetically marked cell groups were deposited into the larval zebrafish brain atlas (mapzebrain.org).

When the complement of cell types was compared between WT and *lakritz* mutants, we could not detect a single-cell type that failed to develop, or one that developed along an altered differentiation pathway. However, we noticed a subtle, yet cumulative, effect of RGC-derived factors on neurogenesis in central visual areas. Without retinal input, progenitors remained on average longer in the G1 phase of the cell cycle. Once having left the cell cycle, daughters transitioned more rapidly into glial precursors, although without fully differentiating into *fabp7a*-positive radial glia. Thus, RGC-derived signals apparently facilitate the transition from uncommitted, cycling progenitors to differentiating neural precursors, without affecting terminal fate selection. A candidate factor is Sonic Hedgehog, whose signaling pathway components are downregulated in *lakritz* mutant brains and which is expressed in developing RGCs across vertebrates[15,16].

Remarkably, synaptic wiring was also apparently unaffected in *lakritz* mutants: When we optogenetically stimulated a pretectal area previously shown to be necessary and sufficient for recognition of prey[45], we could evoke behavior resembling hunting movements[46]. Similarly, OKR-like eye movements could be elicited by photo-stimulating the accessory optic system in the pretectum[47,48]. These findings could not have been predicted from work in other systems. Visual motion, which drives both prey capture and OKR, is the best studied example of how central circuits are shaped by experience and Hebbian plasticity. Experimental manipulation of direction- or orientation-selective RGC inputs disrupts the emergence of properly tuned neurons in cats[12] and ferrets[13]. Even before eye opening, waves of spontaneous activity, which sweep across the retina, simulate sequential activation of neighboring RGCs by directional movement[14,51]; these activity patterns facilitate the formation of visual maps and the segregation of eye-specific inputs in binocular areas[52,53]. In tadpoles, experimentally inverting the direction of optic flow prevents the refinement of the retinotectal map[54]. Perhaps most strikingly, directional tuning of mouse RGCs can be reversed by repeatedly

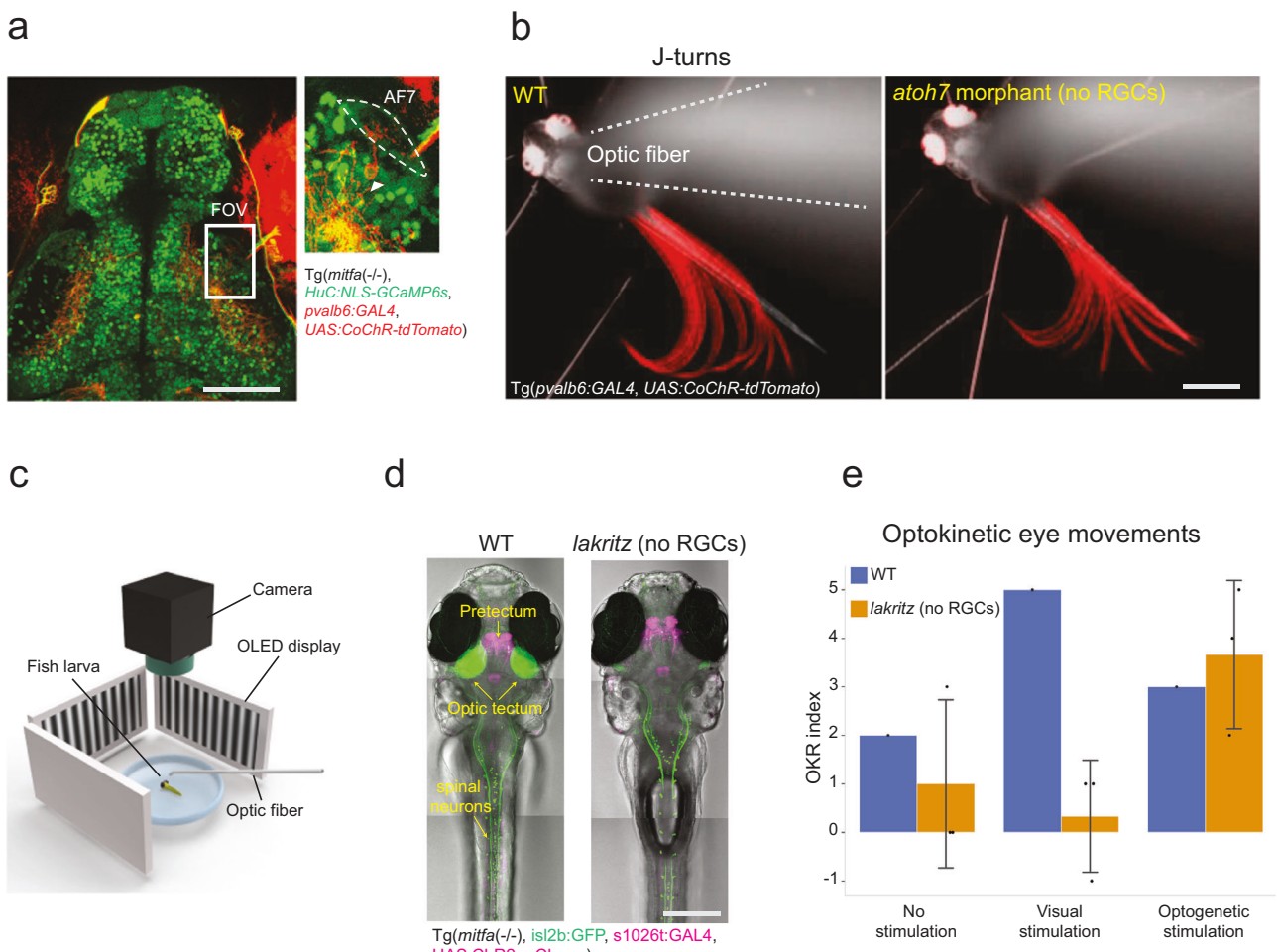

**Fig. 8 | Pretectal circuitry is assembled without retinal input to drive proper behavior. a** Single plane from a triple-transgenic larva carrying *pvalb6:Gal4, UAS:CoChR-tdTomato*, and, for orientation, *elavl3:nlsGCaMP6s* (scale bar = 50 μm). Right panel shows the zoomed-in field of view (FOV) centered on arborization field 7 (AF7). White arrow points to a single, AF7-connected neuron expressing CoChR-tdTomato. **b** Fast recordings of optogenetically induced J-turn-like bouts in larvae with and without RGCs (WT, left; *atoh7* morphant, right). In gray is an average of the larvae before photostimulation (colors are inverted). In red is a standard deviation time-lapse projection of frames containing a J-turn-like bout. Dashed line contours location of optic fiber (scale bar = 500 μm). **c** Illustration of experimental setup. Larvae embedded in agarose on a small transparent dish facing screens showing moving gratings, which generate optic flow. A camera records eye movement, and an optic fiber is positioned above the brain area of interest. **d** Selected images of 6 dpf larvae used for experiment. Maximum z-projection of either WT (left) or *lakritz* (right) larvae. Fish were from the hypopigmented TLN strain (transmitted light) and were transgenic for *isl2b:GFP* (green), *Gal4s1026t* and *UAS:ChR2-mCherry* (magenta). GFP expression was used to phenotype *lakritz* mutants (scale bar = 250 μm). **e** OKR as a function of genotype (*lakritz*, *n* = 3) and stimulus conditions, as indicated. The "OKR index" was calculated by counting the saccades in the expected direction and subtracting the saccades in the opposite direction during the stimulation. For spontaneous eye movements, this index hovers around zero. Increase in "OKR index" indicates a response either to perceived motion (in WT) or to successful photostimulation of the pretectum (in WT and *lakritz*). Data are presented as mean values ± SD. Source data are provided as a Source Data file.

showing a stimulus moving in the initially non-preferred direction[55]. Given that the development of RGCs with different directional selectivity is under tight genetic control[56], these findings imply that correlated sensory activity is, in some circumstances, able to re-program cellular fates. Perhaps unexpectedly, in the face of these reports, the *lakritz* mutant forebrain self-assembles into functional circuits that support visual motion-dependent behavior. It will be interesting to revisit some of the earlier findings, considering that the maturation of synaptic connections may have been delayed due to absence of RGC-dependent factors.

In broad agreement with our findings, recent studies in mouse demonstrated that visual experience is dispensable for development of cortex cell types[18,19], with the exception of neurons in layer L2/3 of V1. Our zebrafish study eliminated all conceivable sources of influence, including retinal waves, synaptic contacts and RGC-derived secreted factors. Our conclusions therefore not only complement previous work in the mouse, but substantially extend it to all sensory-derived factors, to a vertebrate whose brain is still growing, and to subcortical areas that are direct targets of retinal axons. Single-cell transcriptomics, ideally combined in the future with connectomic circuit reconstruction[57] and functional reconstitution via optogenetics, is slated to shed further light on the relative contributions of visual experience and genetic programs to assembly of the brain.

## Methods

### Zebrafish husbandry
Adult and larval zebrafish were maintained on a 14:10 h light:dark cycle at 28 °C. Embryos were kept in Danieau's solution (17 mM NaCl, 2 mM KCl, 0.12 mM $MgSO_4$, 1.8 mM $Ca(NO_3)_2$, 1.5 mM HEPES). For single-cell experiments, one male and one female were placed in individual breeding tanks with a divider the evening before spawning. Dividers were removed at 9.00 a.m. the next morning and fish let spawn until 10.00 a.m. Eggs were collected shortly after and mixed together. All

larvae were reared at a maximum density of 60 individuals in 10 cm petri dish. All animal procedures conformed to the institutional guidelines set by the Max Planck Society, with an animal protocol approved by the regional government (Regierung von Oberbayern).

### Animal genotyping

Larvae carrying *HGn12c:GFP* (short for *Et(HGn12C:GFP)*), or larvae carrying *Gal4s1026t* (short for *Et(fos:Gal4-VP16)s1026t*) and *UAS:GFP* (short for *Tg(UAS:GFP)*), were sorted for GFP expression 24 h after fertilization (hpf). All scRNA-seq experiments were performed at 6 dpf. To obtain *lakritz* mutant and sibling WT larvae, *HGn12C:GFP*-positive *lakritz* heterozygote adults were crossed with adult *lakritz* heterozygotes. To obtain unrelated WT larvae, *HGn12C:GFP* pigmented males or females were crossed with WT TLN, which are homozygous mutant for *mitfa*.

### Cell-dissociation for single-cell RNA sequencing

Six dpf larvae carrying *HGn12C:GFP* (or *Gal4s1026t* X *UAS:GFP*) were used to label neuronal cell types in the forebrain. Ames medium (Sigma A-1420) was used throughout and prepared according to the manufacturer's guidelines. Before animal handling, 300 ml Ames medium was oxygenated in room temperature for 1 h. Buffer pH was adjusted to 7.2–7.3 using HCl and filtered through a standard Steritop filter. Buffer pH was checked again after filtration for a desired range of 7.4–7.5. Oxygenated Ames medium was then placed on ice. Larvae were anesthetized in oxygenated Ames ice slush and rapidly decapitated. Brain material from a maximum of 110 larvae was collected using a broad glass pipette and transferred into chilled oxygenated Ames in a 2 ml tube on ice. Tube medium was replaced with fresh Ames after every material transfer. In addition to transgenic larvae expressing fluorescent protein, 30 non-transgenic larvae were used to adjust FACS gates. Cell suspensions from both samples were prepared in parallel.

Tissue was dissociated into single-cells using a modified protocol[58]. Papain solution [25 U/ml final] was prepared by mixing 4810 μl of oxygenated Ames with 89.3 μl papain stock 42.8 [mgP/ml], 32.7 [U/mgP], 50 μl DNaseI [13 K U/ml] (Sigma D-4527, 40 K Units), and 50 μl L-cysteine [152.2 mM] (Sigma C-1276). Papain solution was then placed for 15 min in a tabletop spinning incubator preheated to 34 °C spinning at 10 RPM. The solution was then examined: though initially milky, the papain solution becomes transparent when activated. Papain solution was not filtered pre- or post-activation. Ames buffer was removed from sample tubes and replaced with 2 ml activated papain solution. Samples were placed in the same tabletop incubator at 34 °C for 1 h spinning at 10 RPM. After 20 min, samples were carefully triturated five times with a narrow glass pipette flamed at the tip to avoid sheering. After 1 h, the sample was placed shortly on a bench to let the biological material sink to the tube's bottom. Papain solution was removed and replaced with 1 ml papain inhibitor solution (4450 μl oxygenated Ames, 500 μl ovomucoid stock, 50 μl DNaseI [13 K U/ml]). 10× ovomucoid stock was prepared as follows: 150 mg BSA (Sigma A-9418), 150 mg ovomucoid (Worthington LS003087), 10 ml Ames buffer; pH adjusted to 7.4, filtered and then stored at −20 °C. After resuspension in inhibitor solution, tissue cells were completely dissociated by triturating a maximum of 30 times with a p1000 pipette (not broad-end) set to 850 μl. A good indicator of a successful dissociation was that there were no observable white particles (brain matter) in solution. Intact eyes were a good indicator that the dissociation was sufficiently gentle to allow high cell-survival. After mechanical dissociation, 1 ml of inhibitor solution was added to each tube. Samples were passed through a pre-wet 30 um filter (Sysmex). Wetting the membrane with Ames buffer allows liquid and small particles to smoothly pass through the mesh filter. Samples were pelleted in a centrifuge pre-cooled to 4 °C for 10 min at 300 × *g*. Supernatant was removed and the pellet resuspended in 2 ml Ames with non-acetylated BSA (4.5 ml oxygenated Ames, 500 μl 4% non-acetylated

BSA, 0.5 μl DNaseI [13 K U/ml]). The resuspended solution was filtered through a pre-wet 20 μm filter into a new 2 ml tube and short spun to get all liquid through the filter. Two microliters of calcein blue [1 μl/ml] was added to the solution to stain live cells. Calcein blue was not added to control samples. Suspensions were kept on ice and processed further by FACS.

### Fluorescence-activated cell sorting (FACS)

BD FACSAria III was used to sort cells. FACS gates were set after 500,000 recorded events to sort live single-cells expressing GFP. Similar gates were used across experiments. Cells were sorted using a 100 μm nozzle (~20 PSI) into 2 ml protein LoBind Eppendorf tubes (Eppendorf 0030108132) placed in a cooling holder. Tubes were treated pre-FACS for 1 h with 2% BSA in Ames while spinning. Before FACS all liquid was removed. The combination of treatment with LoBind plastic ensured cells will not adhere to the collection tube after FACS. Collection tube was filled before cell sorting began with 500 μl FACS collection medium containing: 750 μl oxygenated Ames, 250 μl non-acetylated BSA (stock 4%), 0.1 μl DNaseI [13 K U/ml]. Our FACS gating strategy aimed at collecting live healthy cells labeled by the Tg(HGn12C) transgenic line. To achieve this, we used a series of gates starting with events that showed low side scattering (SSC-A) and high forward scattering (FSC-A). Next, we selected events that showed a linear relationship between the area and height of the forward scattering plot (FSC-H vs. FSC-A). Both, together, allowed us to distinguish intact cells from debris. In the next step we used blue fluorescence from calcein blue (a live-cell stain) and green fluorescence from GFP to collect live cells labeled by the Tg(HGn12C) transgenic line. In total, we aimed to collect 200,000 cells, as in our hands this would fill a 2 ml tube completely. After FACS was completed, cell suspension was centrifuged at 4 °C for 5 min at 300 × *g*. Centrifuge was not pre-cooled. We found it helpful to memorize the tube's orientation, as often it was difficult to visualize the pellet. After centrifugation, the pellet was resuspended in 60 μl of Ames with non-acetylated BSA diluted 1:10 in oxygenated Ames (0.04% non-acetylated BSA, final). The medium was slowly let slide over the pellet multiple times, and then the tube was gently tilted back-and-forth to push the suspension over the location of the pellet a few more times. The cell suspension was placed over ice until used for single-cell sequencing.

### Calculating cell-suspension density

Thirteen microliters oxygenated Ames was supplemented with 2 μl trypan blue (stain for dead cells) and 5 μl cell suspension (1:4 suspension dilution). 20 μl were loaded into a Fuchs-Rosenthal chamber (NanoEnTek, DHC-F01). A minimum of four large squares were counted for live and dead cells using either a dark-field or DIC microscope. Cell density was routinely between 500 and 1000 cells/μl with viability at ~90% (live cells/live + dead cells).

### Single-cell RNA sequencing (scRNA-seq)

Droplet RNA sequencing experiments using the 10X chromium platform were performed according to the manufacturer's instructions with no modifications. Single-chromium chip channels were loaded routinely with 8000 cells aiming to capture 5000 cells with a doublet rate <5%. In our hands, we noticed that loading 8000 cells would usually result in capturing of 3000–3500 cells. For experiments with unrelated WT carrying *HGn12C:GFP*, 17 replicates were collected across 4 experiments. For *lakritz* mutants carrying *HGn12C:GFP*, 10 replicates were collected across 4 experiments. For WT *lakritz* siblings carrying *HGn12C:GFP*, 8 replicates were collected across 3 experiments. For *Gal4s1026t* X *UAS:GFP* larvae, 4 replicates were collected from 1 experiment. The cDNA libraries were sequenced on an Illumina HiSeq 2500 to a depth of ~50,000 reads per cell.

## Alignment of gene expression reads and initial cell filtering

Initial preprocessing was performed using the cellranger software suite (version 3.1.0, 10X Genomics) following standard publisher guidelines. Reads for each channel were aligned to the zebrafish reference genome (GRCz11.98). Further analysis was performed as described below using the Seurat R package[40] on the filtered cellranger output matrices.

## Initial analysis using Seurat

Output from cellranger was loaded into Seurat allowing for 200–4000 genes/cell, 400–8000 UMIs/cell, and a maximum detection of mitochondrial genes of 12% of all transcripts detected in cell. Unless otherwise stated, batch correction was performed using Harmony on experiment and genotype. For analysis of marker genes, the data was first clustered and separated into glutamatergic and GABAergic neuronal datasets, and then batch-corrected and reclustered. For each dataset, we recalculated the 2000 most variable genes ("vst"). We used 18 PCs to cluster our glutamatergic dataset and 20 to cluster our GABAergic dataset. We used a clustering resolution of 1.6 for our glutamatergic dataset and 1.8 for our GABAergic dataset after determining the best resolution using clustree (Zappia et al. GigaScience 2018). Marker genes were extracted using the command FindAllMarkers(…, only.pos = TRUE, logfc.threshold = 0.75), filtered for adjusted $p$ value < 0.05, and inspected individually using the FeaturePlot visualization tool.

Integration of all three genotypes followed the same pipeline, except that in addition to batch correcting using Harmony, we also tested batch correction using the Seurat anchoring method. In short, each combination of experiment and genotype was SCTransformed independently. For each transformed dataset, the 3000 most variable genes were selected as possible integration features. We followed the standard integration pipeline using SelectIntegrationFeatures, PrepSCTIntegration. We used the unrelated WT dataset as our integration space, as it was our largest dataset. We integrated our datasets finally using FindIntegrationAnchors, and IntegrateData. Using this pipeline, we came to the same conclusion as when using Harmony—that we cannot see a difference in cell types between samples.

## Independent clustering of datasets from different genotypes

Data from three genotypes were loaded into Seurat and independently processed and clustered. Separation of the GABAergic and glutamatergic datasets was done as described before, but independently for each genotype. Batch correction was performed for each dataset using Harmony on experimental replicates. For clustering of all cells, 14 PCs were used for all datasets with a resolution of 1. For the glutamatergic dataset, 18 PCs were used with a resolution of 2. For the GABAergic dataset, 22 PCs were used with a resolution of 2.5.

Matching of clusters between datasets was performed based on unrelated WT marker expression identified using the Seurat command: FindAllMarkers(…, only.pos = TRUE, min.pct = 0.25, logfc.threshold = 0.75). For each cluster, we visualized the best marker genes on the UMAP. Upon confirmation that they were descriptive of the cluster (or additional clusters in the case of overclustering), we visualized the expression in the other datasets and assigned all clusters a name based on the expression of a single marker or multiple markers. In the case where no markers could be verified for a cluster, we looked for the nearby clusters on the UMAP, and merged the cluster with another cluster that showed the fewest distinguishing markers.

In some cases, there were clusters we could not assign to all datasets. For the glutamatergic datasets these were: *tubb5/chd4a, nhlh2/zic2a, pvalb6, pax6a, dlx5a, bhlhe23/grm2b, cabp5b, foxb1a/ tfap2e*. For the GABAergic datasets, these were: *aldocab, BX088, adarb2, CR34551, emx2, gata2a/nxph1, gyg1b, irx1b, crhbp, otp, onecut1, penkb, pnocb, txn/nr4a2a, uncx4.1*. These differences were often the result of the larger cell numbers in the unrelated WT

dataset, allowing for the clustering algorithm to more accurately separate cell types. In the cases where a cell type is a rare one, the cells were often merged with larger nearby clusters. Some notable exceptions: *adarb2* cells clustered strongly in *lakritz* mutants and siblings, but not in the unrelated WT dataset; *otp*, neurons, which are mainly glutamatergic, appeared in the unrelated WT GABAergic dataset; *uncx4.1* formed a significant cluster in the unrelated WT dataset, but was largely missing from the *lakritz* and sibling datasets, as was the same for the glutamatergic *dlx5a* cluster. However, there was never a case where cells expressing these markers were completely missing from all other datasets. For closer inspection, see Figs. S4–S7.

## Determination of confounding batch effects

WT and *lakritz* datasets were first analyzed independently. To determine whether there are batch effects within each dataset across experiments, each dataset was first analyzed independently. Normalization, variable feature selection, scaling, and principal component analysis were performed with either WT or *lakritz* mutant cells. The DimPlot visualization tools (Seurat) was used to inspect different combinations of the first 5 PCs where cells were color-coded by experiment. We could not detect a batch effect in the *lakritz* dataset. For the WT dataset, where we could detect a batch effect, we tested whether Harmony can correct the effect. To test whether WT and *lakritz* datasets were integrated properly also in absence of batch correction, we followed standard procedure, but omitted a batch-correction step.

To test whether *lakritz* cluster structure is preserved following integration with WT cells, we first clustered the *lakritz* data independently and saved the results as a large matrix containing cell and "*lakritz*-only" cluster identities. We then followed standard procedure to cluster WT and *lakritz* cells together. Lastly, we plotted *lakritz* cells according to their new UMAP coordinates (embedded with WT) and color-coded the cells according to their "*lakritz*-only" clusters.

## In-silico cell-type ablation analysis

Cell identity and cluster information were imported from a completed clustering analysis. Iteratively, *lakritz* cells belonging to a single cluster were removed from the count matrix. After, removal of the cells, the matrix was processed following the standard Seurat pipeline without batch correction. To determine whether we could observe an area missing in *lakritz* cells, we visualized all three genotypes together in UMAP space. Upon determination of a missing area, we opened an image where cells in the cluster missing *lakritz* cells were highlighted. We applied this analysis separately to datasets containing glutamatergic or GABAergic neurons for all clusters. We avoided any type of batch correction as that can mask differences.

Overall, visual inspection correctly identified a missing cluster in *lakritz* in 87% of glutamatergic clusters and 71% of GABAergic clusters (Supplementary Figs. 9a–c and 10a–c). Together these clusters contain 90% of all cells we sequenced. In cases of clusters where we could not detect an effect, we noticed that these clusters often embedded poorly in a 2D UMAP; they show a dispersed pattern, apparently mixed among many other clusters, making it difficult to visualize these cells as a single cluster (Supplementary Figs. 9d and 10d).

## Alignment of PCs between genotypes

Datasets were batch-corrected using Seurat's anchoring pipeline as described before. PCs were calculated using the Seurat RunPCA function for the first 15 PCs. For each comparison, only genes that appeared across PCs in both groups were compared. PCs were compared by calculating the cosine similarity between all PCs. PCs were compared by calculating the cosine similarity and extracting the highest value for each PC, generating a distribution of values for most similar PCs. We generated our null distribution, by randomly dividing

our WT dataset into two groups similar in ratio to *lakritz*/unrelated WT. We repeated this ten times to generate a null distribution for PC similarity across different datasets. We used a Wilcoxon signed-rank test to compare the different distribution. The small variation in gene expression had a marginal effect on the principal components (PCs). While, in some cases, PCs are ranked differently, the top PCs are identical across groups (Supplementary Fig. 11a, b).

### Comparing neighborhood embedding between genotypes

If the expression of individual genes were systematically different between genotypes then the nearest neighbors of *lakritz* cells should more often be other *lakritz* cells rather than a mix of both. This statistical effect should be enhanced if the differentially expressed genes are also marker genes and would create local 'hotspots' in the UMAP space, populated preferentially by either *lakritz* or WT cells. Alternatively, changes in gene expression might be more distributed, with most, or all, clusters affected similarly.

For each cell, the neighborhood was determined by extracting the 19 nearest neighbors' genotype composition and calculating the cosine distance between the observed neighborhood genotype composition (unrelated WT/sibling WT/*lakritz*) and the expected neighborhood genotype composition. The null distribution was generated by randomly shuffling the genotype labels and recalculating the distance between observed and expected neighborhood genotype composition. From this, we extracted the standard deviation and calculated a Z-score for each cell in the original dataset. We color-coded our UMAP based on the Z-score to uncover areas in the UMAP (potentially clusters) that show significantly altered neighborhoods in *lakritz*.

We found that altered neighborhoods exist throughout the *lakritz* dataset (Supplementary Fig. 11c), suggesting that the differences are broadly distributed among the clusters. The magnitude of overall changes between pairs were highly similar, i.e., unrelated WT clusters were not more similar to WT siblings than to mutants, inconsistent with the absence of RGCs causing global or local drifts in gene expression.

### Analysis of globally differentially expressed genes between genotypes

In the absence of a generally agreed-on statistical test for differences in these kinds of multidimensional datasets, we carried out a straightforward three-way comparison between mutants, their WT siblings, and unrelated WT. For analysis of genes differentially expressed globally between genotypes, datasets were batch-corrected using Seurat's anchoring pipeline as described before. We used the FindMarkers command with default parameters comparing either unrelated WT with *lakritz* or unrelated WT with sibling WT. Globally differentially expressed genes were filtered using adjusted $p$ values <0.05. We highlighted genes for which the log fold change was either larger than 1 or lower than −1, or where the ratio of expressing cells across groups was either higher than 2 or lower than 0.5. This revealed that few of the potentially dysregulated genes pass a threshold defining cell-type markers. We also ran this analysis using Harmony for batch correction, but this led to no differentially expressed genes.

### Finding clusters with altered transcriptomes

The analysis was applied as reported before[35] on datasets batch-corrected using Seurat's anchoring points pipeline as reported before. In short, for each cluster we measured the ratio between WT and *lakritz* mutants for the set of marker genes and for all genes detected in the cluster. We then compared these two distributions using a Wilcoxon signed-rank test and corrected the $p$ values for multiple testing (depending on the number of clusters) using Bonferroni's correction. We extracted the WT identities belonging to the cluster and projected them into the formerly analyzed WT dataset in which we performed our marker analysis. We color-coded the cells based on their p-value for

transcriptome alteration. From this, we extracted a list of markers expressed in clusters predicted to be the most altered in *lakritz* mutants compared to WT. We performed this analysis also by applying the same processing pipeline starting with only glutamatergic or GABAergic cells essentially applying normalization within the separated dataset. We also tested this pipeline following Harmony batch correction.

### Comparing cell-type proportions

For each experiment, we calculated the relative proportion of cells in each cluster as a fraction of the total number of cells captured across replicates in a single experiment. We compared the distribution of proportion of each cluster between WT and *lakritz* mutants using a Wilcoxon signed-rank test and corrected the p-values for multiple testing using Bonferroni's correction based on the number of clusters.

### Developmental trajectory analysis

The identities of cells belonging to the neural precursor cluster (expressing *ngn1* and *ascl1b*) were extracted and highlighted. Marker genes for this cluster were calculated using the FindMarkers Seurat function. Each gene was visually inspected using the FeaturePlot visualization tool to determine whether it is expressed only in the cluster (transient expression), or also in other clusters (continuous expression). For each gene we also summarized whether it is expressed in other cell classes (GABAergic neurons, glutamatergic neurons, habenula neurons, and progenitors).

### Trajectory inference analysis

Trajectory inference was performed in Python. Data were processed using the scanpy[59] package before calculating RNA velocities using the ScVelo[38] package. Trajectory inference was supplemented using the cellrank[39] package. Inference of intronic vs. exonic reads was performed on Linux server using the velocyto[38] package with default parameters. Cell-cycle analysis was performed using scanpy with gene sets established for G2M-phase and S-phase scoring[40,60]. Gene names were altered to match zebrafish nomenclature. Python gene lists were created in python as follows:

s_genes_zebrafish = ["mcm5", "pcna", "tyms", "fen1", "mcm2", "mcm4", "rrm1", "unga", "ungb", "gins2", "mcm6", "cdca7a", "cdca7b", "dtl", "prim1", "uhrf1", "cenpu", "hells", "rfc2", "rpa2", "nasp", "rad51ap1", "gmnn", "wdr76", "slbp", "slbp2", "ccne2", "ubr7", "pold3", "msh2", "atad2", "atad2b", "rad51", "rad51b", "rad51c", "rad51d", "rrm2", "rrm2b", "cdc45", "cdc6", "exo1", "tipin", "dscc1", "blm", "casp8ap2", "usp1", "pola1", "chaf1b", "brip1", "e2f8"]

g2m_genes_zebrafish = ["hmgb2a", "hmgb2b", "cdk1", "nusap1", "ube2c", "birc5a", "birc5b", "tpx2", "top2a", "ndc80", "cks2", "nuf2", "cks1b", "mki67", "tmpoa", "tmpob", "cenpf", "tacc3", "smc4", "ccnb2", "ckap2l", "aurkb", "bub1", "kif11", "anp32e", "tubb4b", "gtse1", "kif20ba", "cdca3", "jpt1a", "cdc20", "ttk", "cdc25b", "kif2c", "rangap1a", "rangap1b", "ncapd2", "dlgap5", "cdca8", "ect2", "kif23", "hmmr", "aurka", "anln", "lbr", "ckap5", "cenpe", "ctcf", "nek2", "g2e3", "gas2l3", "cbx5"]

All analysis pipelines can be found in online tutorials:

Scanpy: https://scanpy-tutorials.readthedocs.io/en/latest/pbmc3k.html

Cell-cycle analysis:

https://nbviewer.org/github/scverse/scanpy_usage/blob/master/180209_cell_cycle/cell_cycle.ipynb

RNA velocity: https://scvelo.readthedocs.io/en/stable/VelocityBasics/

Cellrank: https://cellrank.readthedocs.io/en/stable/cellrank_basics.html

### HCR fluorescent in situ RNA labeling (HCR-FISH)

HCR stains were performed according to the manufacturer's instructions (Molecular Instruments) with no modifications. Larvae used for HCR-FISH were reared in PTU (1-phenyl 2-thiourea) from 24 hpf to 6

dpf. At 6 dpf, larvae were fixed following instructions. All larvae were *HGn12C:GFP*-positive and stained for a maximum of two transcripts labeled with either or both Alexa546 and Alexa647. All probes were purchased from the manufacturer (Molecular Instruments). Imaging was performed on a commercial Zeiss confocal (LSM780).

For WT and *lakritz* comparisons, larvae were generated from crosses of *lakritz* heterozygous carriers, one of which was double-transgenic for *HGn12C:GFP* and *isl2b:RFP* (*Tg(−17.6isl2b:TagRFP)^{zc80tg}*). Homozygous *lakritz* mutants were identified based on the absence of RFP signal in the tectum. Comparative stains were performed only on siblings from the same clutch stained in parallel for each gene. Gene expression was compared in *lakritz* mutants and WT using identical confocal settings in both samples. Images were collected from a minimum of two replicates for each condition. For each individual either one or two tiles were imaged.

### Morphological registration using ANTs

Brain registration using ANTs was performed as described before[30]. In short, a standard brain was generated using 12 HCR-stained *HGn12C:GFP*-positive larvae using ANTs. All HCR-FISH stains were registered to the *HGn12C:GFP* standard using the GFP signal as a reference. In case of drift during image acquisition, the transmitted-light channel was used to correct drift using the MultiStackReg ImageJ plugin. Once the affine transformation was saved, it was applied to all other channels.

To compare WT and *lakritz* stains, individual confocal stacks were registered first to the *HGn12C:GFP* standard, and then differences were inspected by eye. In cases where differences were detected, the original stacks were inspected to validate differences.

Anatomical masks from the mapzebrain atlas (mapzebrain.org) were registered to the *HGn12C:GFP* standard using a bridging *Tg(elavl3:H2B-RFP)*[61] reference from the atlas. Double-transgenic *Tg(elavl3:H2B-RFP)* and *HGn12C:GFP* larvae were independently registered to the *HGn12C:GFP* standard to generate an H2B-RFP stain in the standard atlas space. Transgenic fluorescence was successfully preserved in *Tg(elavl3:H2B-RFP)* by modifying the HCR fixation protocol as reported before[62].

To track the development of HCR-labeled gene expression at different ages (Table S4), we generated an average standard brain for 3 and 4 dpf. To align individual stains, we first aligned them to their age-matched standard brain and then iteratively aligned them to the older standard brains.

### HCR-FISH image analysis

Individual confocal stacks of HCR stains were aligned to the standard *HGn12C:GFP* expression pattern as described before. For analysis of HCR-FISH signals, individual confocal stacks were binarized using the ImageJ "RenyiEntropy" algorithm applied to individual slices using each slice's histogram. In slices where the algorithm produced sudden spikes in the total number of pixels compared to surrounding slices, pixels were instead binarized using the "maxEntropy" algorithm. This step improved the binarization process where the "RenyiEntropy" algorithm was sensitive to sudden changes in noise. However, both algorithms were sensitive to different kinds of noise which allowed to generate a combined smooth binarization mask. The binarized pixels were then multiplied by the image intensity in each pixel. The morphological signal analysis was performed on the product of these images. Registered anatomical masks were generated as described before. For total expression analysis, the sum of the intensity across the binarized pixels was calculated for each anatomical mask. This analysis was performed for replicates of the same stain and averaged across replicates. The final heat matrix was calculated by normalizing the expression of each gene to the highest value for the specific gene. For background normalization, the background was defined as the signal intensity in pixels inside the anatomical mask that are not HCR positive binarized pixels.

For predictions of best matches between brain areas and gene expression, the background-normalized signal values were normalized in the heatmap to the highest value in each brain region.

For analysis of genes involved in neuronal differentiation (Tables S3 and S4), HCR-FISH signals were quantified in each of the brain areas described in the table and normalized to the GFP signal in the *HGn12C:GFP* background stain.

### Optogenetic stimulation of pretectum and behavioral tracking

Optogenetic stimulation and behavioral tracking were performed as reported before[48]. In short, after embedding and removal of agarose close to the eyes, an optic fiber was placed on top of the fish head, targeting the pretectum. In order to track the eyes while in dark, larvae were illuminated from the bottom using an 850 nm infrared LED. Light emitted by optogenetic illumination was filtered out by an IR filter in front of the recording camera (Thorlabs absorptive filter, ND = 1.0). For focal optogenetic activation with ChR2, a 50 μm optic fiber (AFS50/125Y, Thorlabs) was used to shine blue light (473 nm, 20–40 mW/mm², Omicron Lighthub) onto the right or left pretectum. In each experiment, larvae were presented with a phase of stationary gratings, followed by moving gratings (40 s), followed by stationary gratings, followed by blue light illumination (60 s) and stationary gratings.

For behavioral tracking of prey capture-associated tail movements, video capture and optogenetic activation were performed as described before[44,46]. Larvae were injected with *atoh7* morpholinos (Gene Tools, LLC) at the one-cell stage. The *atoh7* gene is disrupted in *lakritz* mutants, and *atoh7* morpholinos faithfully phenocopy the *lakritz* mutation at embryonic and early larval stages[63]. Larvae expressed the *isl2b:GFP* transgene (*Tg(−17.6isl2b:GFP)^{zc7Tg}*), which allowed sorting of *atoh7* morphants. Larvae were also transgenic for *Tg(elavl3:nlsGCaMP6s)* for overall orientation and targeting of the optic fiber. Larvae were selected for complete absence of GFP from pretectum and tectum prior to experiments. Analysis of J-turns was performed by inspecting larval bouts and manually annotating the onset of bouts classified as J-turns.

For optogenetic activation of pretectal OKR circuitry, homozygous *lakritz* mutant larvae carrying *isl2b:GFP* (*Tg(−17.6isl2b:GFP)^{zc7tg}*), *Gal4s1026t*, and *UAS:ChR2-mCherry* (*Tg(UAS:ChR2(H134R)-mCherry) s1985t*) from the TLN strain were used. As controls, wildtype larvae from the same clutch (lacking mCherry signal) expression were used. Larvae were identified as *lakritz* mutants by the absence of GFP-labeled retinal projections. To track the OKR, the angle of each eye was calculated relative to the body midline. During visual stimulation (gratings moving), the eyes of a fish almost exclusively saccade in a single direction intermittent with a smooth movement in direction of motion. When there is no visual stimulation (stationary gratings), the eyes of a fish will saccade in one direction and then in the opposite direction. Hence, saccades are a reliable readout of the OKR. To calculate an OKR index, the saccades in one direction were subtracted from the saccades in the other direction for each eye – producing higher values during OKR. The average of both eyes was taken as the OKR index.

### Validation of *lakritz* genotype

For larvae, *lakritz* mutants were routinely identified by their dark pigmentation, when possible. When TLN (*mitfa*-/-) or PTU (1-phenyl 2-thiourea) treated larvae needed to be identified, either *isl2b:RFP* (*Tg(−17.6isl2b:TagRFP)^{zc80tg}*) or *isl2b:GFP* (*Tg(−17.6isl2b:GFP)^{zc7tg}*) were used. The *isl2b* promoter/enhancer drives transgene expression in RGCs, trigeminal ganglia, and spinal neurons. In *lakritz* mutants, there are no RGCs, but expression in trigeminal ganglia and spinal neurons is unaffected. In adults, fin clips were used to determine carriers of the *lakritz* mutation as described[28]. In short, myTaq Extract-PCR kit (Meridian Bioscience) was used to amplify a 300 bp fraction of the *atoh7* gene containing the *lakritz* mutation using the following primers: *Fw_ccggaattacatcccaagaac, Rv_ ggccatgatgtagctcagag*. The amplified

product was digested using the StuI restriction enzyme over night at 37 °C. Carriers of the *lakritz* mutation show three products on an agarose gel at 300, 200, and 100 bp. WT fish show two products at: 200 and 100 bp.

**Reporting summary**

Further information on research design is available in the Nature Portfolio Reporting Summary linked to this article.

## Data availability

Relevant data supporting the key findings of this study are its Supplementary Information files or from the corresponding author upon request. All raw and processed scRNA-seq datasets reported in this study have been deposited in NCBI's Gene Expression Omnibus (GEO) under the accession number GSE238240. Source data are provided with this paper.

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

## Acknowledgements

We are grateful to the Max Planck Campus Martinsried sequencing facility for their assistance, especially Markus Oster for FACS processing and troubleshooting and Marja Driessen and Rin Ho Kim for next-generation sequencing and initial bioinformatics. We would also like to thank Yvonne Kölsch for help in sample preparation. We thank Gregory Marquart, Manuel Stemmer, Inbal Shainer, Johannes Larsch, Karthik Shekhar, Joshua Hahn, Salwan Butrus, Christian Mayer and Wolfgang Enard for critical feedback and discussions. This work was supported by the Max Planck Society.

## Author contributions

S.S. and H.B. conceived the study. S.S performed experiments and analysis. M.W.S performed optogenetics experiments. I.A.-A. provided technical assistance. S.S. and H.B. wrote the manuscript and prepared the figures. K.K. provided the enhancer-trap line HGn12c:GFP.

## Funding

## Competing interests

The authors declare no competing interests.
