## [Peer Review File · Nature Communications]

Retina-derived signals control pace of neurogenesis in visual brain areas but not circuit assemblyREVIEWER COMMENTS

Reviewer #1 (Remarks to the Author):

In this study from the Baier lab, the authors characterized visual circuit components (namely cell types) in the thalamus and pretectum of zebrafish larvae, and investigated the differences when animals are raised without retinal ganglion cells (RGCs) in lakritz mutants. The authors used single cell transcriptomics to characterize Glutamatergic and GABAergic cell types in the thalamus and pretectum, and verify these findings further with HCR method. Using this approach, the authors showed that the difference of cell types in lakritz mutants without RGC inputs, is very small, and argue that the development of pretectum and thalamus is largely unaltered in lakritz mutants, except the effect that they observed in the transition from neural progenitors to recently differentiated neurons. Finally, the authors also showed that stimulating distinct components of pretectum in lakritz mutants elicited proper visually evoked behaviors such as J-turns and optokinetic response.

This is an exciting study with lots of different angles about the contribution of visual inputs in the generation of cell types, brain development and its consequence for animal behavior. In fact, I have reviewed this manuscript earlier for another journal, and I am very happy to see that the authors took all my major feedback (about new analysis, experimentation and interpretation of their results) very seriously, and did their absolutely best to address them in this revised version. I therefore think that this revised manuscript is a substantially improved version of the original manuscript in multiple ways. I therefore strongly support the publication of this revised manuscript in Nature Communications. As I go over the revised manuscript, I notice few points about the new analysis and data that the authors added, and some of the impact of the data in authors interpretations. I will try to raise these points in below sections. I also try to put my feedback below in the context of my earlier comments, to highlight how much the authors have done to address my earlier comments, and how to interpret these new results within the main message of this manuscript.

Comments:

- 1) In my comments to previous version of this manuscript, I mentioned that several arguments of the authors on the lack of major reorganization of diencephalic cell types relies on comparing UMAP distributions. And I asked "How quantitative really are these comparisons, and what is the limit of this approach to visualize and quantify small differences in cell types, their distributions, their organization?" I see that the authors now successfully addressed this major point, and in several places they had to invent new and convincing methods/analysis to strengthen their claims. However, in few places I felt like the newly introduced analysis can be explained in concise form (1-2 sentence) in the main result section, in addition to the details in the method section. This will make it easier for the readers to go over the text. I can see for example Line 214 "we applied an analysis used before to uncover such clusters" is a good place to add few sentences about this interesting method.
- 2) In my earlier comments I wrote "Since the authors have very good HCR stains for several of their interesting marker genes, wouldn't be wise to support their claims also with better anatomical and also spatial data?". I see that authors took this very seriously and did a series of important additional HCR

stains, which increased the quality of this manuscript. However I also see that some of these HCR stains in Supp Figure 13 highlight the fact that the content of some neural populations are altered in lakritz mutants, is it not ? Does this also mean that perhaps the analysis/comparison in Figure 4 is not able to capture such fine changes in neural populations ? How do these results relate the authors rather strong claim about the idea that the neural populations are not changes in lakritz mutants ? I do not argue that there are big changes in Lakrtiz mutants but this HCR data in Supp Figure 13, also suggests that there are changes in some neural populations, and this might need some slight toning down for the claims in the discussion, and perhaps even abstract. This will also help accommodating these new and exciting results that some populations are more sensitive to the lack of RGC information/inputs, and some not, quite like the mouse Layer 2/3 results the authors cited.

3) Also It is not clear how the gene sets for the HCR in Supp Fig 13 is selected ? I assume based on the analysis in Sup Fig 12 ? It will be great if you could explain this gene selection for Supp Figure 13 HCRs a bit clearer in the text or show some figures about this selection.

4) Earlier version of the manuscript, I commented “I understand that the authors showed few distinct gene expression patterns, but how does over all organization of pretectal/thalamic circuits are altered? Given the brain atlas technology of that authors, how difficult would it be to better quantify the cellular diversity and anatomical organization of these regions, and compare to lakrtiz animals ?” I see that authors successful integrate this point in the updated manuscript. Can these HCR images, which are now aligned to “mapZebbrain”, be used for additional quantification for identifying if any population is altered in lakritz mutants ? If the authors thinks that showing these HCR stains is sufficient since the differences are big enough, I am fine with this. However, if the argument is that there are no changes in lakrtiz mutant neural populations, then perhaps some image quantification could help.

5) The new analysis on the progenitors that the authors added in this revised manuscript is very exciting with very interesting findings about the arrest of progenitor cell states in the lakritz mutants. This is also in line with my earlier comment on “diving deeper into this alterations of neural progenitors”. And I see that authors did a lot of extra work to dive deeper into the mechanisms of this altered progenitors states, by using RNA velocity analysis and additional plots. Since I am not an expert in such RNA velocity analysis, it is difficult for me to judge the method used to generate these RNA velocity maps.

6) Line 191 “A large number of genes indeed varied between lakritz and WT” I could not find the figure associated with this statement.

7) Line 195, “While, in some cases, PCs are ranked differently, the top PCs are identical across groups (Supplementary Figs. 11a,b)” given the fact that these top PCs explain only the big variances, is this a good method really to use in order to say there are no changes in expression patterns ? I don’t have strong opinion about this firtue, but I am not sure how strong this figure adds to the support of the major claim

Reviewer #2 (Remarks to the Author):

The manuscript by Sherman and colleagues is an interesting study investigating the question how a sensory part of the brain develops in the absence of sensual input. They picked an interesting system, namely the zebrafish lakritz mutant, which does not develop retinal ganglion cells and hence retinorecipient areas of the brain develop devoid of external input. The authors used a sophisticated and exhaustive transcriptional profiling. Surprisingly they found very little transcriptional changes among the 77 neuronal types they identified. A few individual genes were slightly misregulated, but the neuronal types developed in normal proportions and locations. Somewhat akin to the prolongation of critical periods during sensual deprivation, cell cycles were slower and terminal differentiation in the brain is slowed down.

Although the result is not completely surprising, given the apparent normal development of activity blocked zebrafish embryos in classical experiments, the extent of normal neural circuit assembly is astonishing. A nice icing of the cake is the final experiment, where the authors optogenetically stimulate pretectal neurons to evoke proper eye movement in the absence of visual input.

I only have few minor comments that the authors may consider. It is not clear to the reviewer why in the J-turn experiment morphants instead of mutants (or crispants) have been used.

In Figure 7e, a dot plot would have been more informative than a box plot.

Overall this is a very interesting paper with extensive and sensible use of data analysis of single transcriptome data, beautiful anatomical information, and a very nice behavioral confirmation that neural circuit formation is independent of sensory input.

Reviewer #3 (Remarks to the Author):

The manuscript of Sherman et al. demonstrates that inputs and signals from zebrafish retina are not necessary for the establishment of neuron subtypes (molecular identities) and functional synaptic connections in the brain areas which are involved in visual processing and responses. This is a significant and original contribution to the fundamental question of to what extent brain maturation is shaped by external stimuli.

The model organism is a zebrafish lakritz mutant (*atoh7^{-/-}*) that lacks retinal connections, established in a laboratory of the senior author over 20 years ago and well characterised. Single cell RNA-seq followed by a complex bioinformatics analysis convincingly show that developmental trajectories and diversity of neurons were not affected. This agrees with recent data on mice, and extends the conclusion from this previous study, because it excludes the potential role of factors secreted from retinal connections.

Furthermore, optogenetic stimulation of pretectal areas evoked normal visual behaviours, credibly demonstrating that central visual circuits developed normally and were functional. This result is unexpected and exciting because previous research, mainly on mammals, showed that a disruption of visual input alters the formation of visual maps and wiring of the visual system. Overall, I find this study very interesting, especially for neurodevelopment researchers. Additionally, transcriptomic data with spatial resolution, which are a part of this study, provide very useful information about markers which can be used to distinguish neuronal sub-populations in zebrafish, and possibly also in other vertebrates. The manuscript is generally well written. The title is adequate, the Abstract is a good summary of the

study, the Introduction is concise and informative, interpretations and conclusions in the Discussion are clear and justified. My concerns (major flaws) are directed mainly to the structure of the Result section.

Major flaws

1) In Results, I see a problem with an overload with bioinformatics data. Three chapters (lines 140-225) correspond to figures 3 and 4. While these two figures are barely mentioned, other twelve (supplementary) figures are quite detailed, discouraging further reading. Actually, one of these chapters cites only supplementary figures. If these results were not crucial enough to be in the main text, why devote so large an amount of space to them? I understand that all these impressive bioinformatics analyses were very complex and necessary to draw strong conclusions, but the majority of readers probably would not follow them. I wonder if it would be possible to move some of these descriptions to Methods or to the supplement, not to overshadow the message with results which are supportive. I also think that Fig. 3 a-b could be moved to supplements, Fig 3c-d could be merged with Fig. 4, and these results could be compactly described in one chapter with one bigger message. Similar situation is with Fig. 5 and 6. Are the panels 6a, b, c, d necessary for the main text? Could the two chapters corresponding to figures 5 and 6 be merged into one concise chapter?

2) The chapter "Expression of progenitor and early precursor..." cites only supplementary tables. Moreover, it describes a panel of in situ hybridisations, but does not show any of these stainings. They are only summarised in the tables in the form of numbers from statistical analysis. Representative images should be provided. Also, the quantitative analysis, though well done from the methodological point of view, does not convince me, because the differences are usually small, up to 1.5-fold change. I am sceptical if in situ hybridisation can show such differences reliably. In my opinion these results raise doubts rather than reinforce conclusions from RNA-seq. I would rather not place them in this paper.

3) The conclusions about cell cycle exit delay are based solely on single cell RNA-seq analysis. I suggest using another method to confirm this claim. Is it possible to analyse cell cycle by a pulse BrdU incorporation combined with in situ hybridisation or FACS?

Minor flaws

1) Line 80-81: "Supplementary Figs. 1b-f" – I suppose it was to be "Figs 1b-f"

2) Fig. 1 legend – to make this figure+legend self-explanatory, I suggest including the information that green cells were sorted for subsequent single-cell RNA-seq

3) Fig. 1 g-h – would it be possible to mark which cell cluster belongs to which brain part? for example in the x-axis.

4) Fig. 2 looks a bit messy. Do you need figure 2b, which is a repetition of fig. 1a? Why GABAergic clusters are not shown for 2a and 2f? Why there are no gene labels in the right panel, unlike the left and medial panels?

5) PTU abbreviation should be explained.

6) Information that "all probes were purchased from the manufacturer" should be supplemented with at least the name of the manufacturer.

Loose suggestion

1) While the first chapters of Introduction are excellent, the last Introduction chapter, which practically describes the results, is weaker. I find it too long, and at the same time it lacks a clear question. I suppose it could be easily improved.

2) The Discussion section lacks a closure. The last sentence serves this purpose, but one paragraph with a couple of sentences would do the job better

RESPONSE TO REVIEWER COMMENTS

Reviewer #1:

In this study from the Baier lab, the authors characterized visual circuit components (namely cell types) in the thalamus and pretectum of zebrafish larvae, and investigated the differences when animals are raised without retinal ganglion cells (RGCs) in lakritz mutants. The authors used single cell transcriptomics to characterize Glutamatergic and GABAergic cell types in the thalamus and pretectum, and verify these findings further with HCR method. Using this approach, the authors showed that the difference of cell types in lakritz mutants without RGC inputs, is very small, and argue that the development of pretectum and thalamus is largely unaltered in lakritz mutants, except the effect that they observed in the transition from neural progenitors to recently differentiated neurons. Finally, the authors also showed that stimulating distinct components of pretectum in lakritz mutants elicited proper visually evoked behaviors such as J-turns and optokinetic response.

This is an exciting study with lots of different angles about the contribution of visual inputs in the generation of cell types, brain development and its consequence for animal behavior. In fact, I have reviewed this manuscript earlier for another journal, and I am very happy to see that the authors took all my major feedback (about new analysis, experimentation and interpretation of their results) very seriously, and did their absolutely best to address them in this revised version. I therefore think that this revised manuscript is a substantially improved version of the original manuscript in multiple ways. I therefore strongly support the publication of this revised manuscript in Nature Communications. As I go over the revised manuscript, I notice few points about the new analysis and data that the authors added, and some of the impact of the data in authors interpretations. I will try to raise these points in below sections. I also try to put my feedback below in the context of my earlier comments, to highlight how much the authors have done to address my earlier comments, and how to interpret these new results within the main message of this manuscript.

Comments:

1) In my comments to previous version of this manuscript, I mentioned that several arguments of the authors on the lack of major reorganization of diencephalic cell types relies on comparing UMAP distributions. And I asked "How quantitative really are these comparisons, and what is the limit of this approach to visualize and quantify small differences in cell types, their distributions, their organization?" I see that the authors now successfully addressed this major point, and in several places they had to invent new and convincing methods/analysis to strengthen their claims. However, in few places I felt like the newly introduced analysis can be explained in concise form (1-2 sentence) in the main result section, in addition to the details in the method section. This will make it easier for the readers to go over the text. I can see for example Line 214 "we applied an analysis used before to uncover such clusters" is a good place to add few sentences about this interesting method.

We have now added text to better explain our methods, also to highlight innovations.

2) In my earlier comments I wrote “Since the authors have very good HCR stains for several of their interesting marker genes, wouldn’t be wise to support their claims also with better anatomical and also spatial data?”. I see that authors took this very seriously and did a series of important additional HCR stains, which increased the quality of this manuscript.

However I also see that some of these HCR stains in Supp Figure 13 highlight the fact that the content of some neural populations are altered in *lakritz* mutants, is it not? Does this also mean that perhaps the analysis/comparison in Figure 4 is not able to capture such fine changes in neural populations? How do these results relate the authors rather strong claim about the idea that the neural populations are not changes in *lakritz* mutants? I do not argue that there are big changes in *Lakritz* mutants but this HCR data in Supp Figure 13, also suggests that there are changes in some neural populations, and this might need some slight toning down for the claims in the discussion, and perhaps even abstract. This will also help accommodating these new and exciting results that some populations are more sensitive to the lack of RGC information/inputs, and some not, quite like the mouse Layer 2/3 results the authors cited.

It is possible that some neuronal populations are altered in subtle or not-so-subtle ways, and we point out detectable differences at several places in the manuscript. Still our main conclusion stands, namely that the development of cell types is unaltered in *lakritz* mutants. While we do see lower expression of some transcripts, including calcium binding proteins, the specific cell populations appear in the correct anatomical location. We do not call into question the well-supported observation that genes are regulated by neural activity. In fact, we believe we exercised sufficient caution in how we framed certain findings (e. g., “largely unaltered”, “most, if not all,..”). Similarly, in the Discussion we emphasize not an absolute finding but one where we “could not detect a difference”. Please also note Reviewer 3’s comment that our “interpretations and conclusions are clear and justified”.

3) Also It is not clear how the gene sets for the HCR in Supp Fig 13 is selected? I assume based on the analysis in Sup Fig 12? It will be great if you could explain this gene selection for Supp Figure 13 HCRs a bit clearer in the text or show some figures about this selection.

We revised the following sentence to clarify how genes were selected for HCR comparisons: “We then compared the expression patterns of the top cluster-specific markers in WT and *lakritz* mutants by HCR in-situ labelings.”

4) Earlier version of the manuscript, I commented “I understand that the authors showed few distinct gene expression patterns, but how does over all organization of pretectal/thalamic circuits are altered? Given the brain atlas technology of that authors, how difficult would it be to better quantify the cellular diversity and anatomical organization of these regions, and compare to *lakritz* animals?” I see that authors successful integrate this point in the updated manuscript. Can these HCR images, which are now aligned to “mapZebbrain”, be used for additional quantification for identifying if any population is altered in *lakritz* mutants? If the authors thinks that showing these HCR stains is sufficient since the differences are big enough, I am fine with

this. However, if the argument is that there are no changes in lakritz mutant neural populations, then perhaps some image quantification could help.

We registered the WT HCR data to the standard brain of the digital atlas, but did not do this for the *lakritz* mutant, because the image processing/alignment/averaging procedure does not lend itself easily to quantification of expression levels. Also, the observed effect sizes were big enough to be detected by visual inspection of individual specimens.

*5) The new analysis on the progenitors that the authors added in this revised manuscript is very exciting with very interesting findings about the arrest of progenitor cell states in the *lakritz* mutants. This is also in line with my earlier comment on “diving deeper into this alterations of neural progenitors”. And I see that authors did a lot of extra work to dive deeper into the mechanisms of this altered progenitors states, by using RNA velocity analysis and additional plots. Since I am not an expert in such RNA velocity analysis, it is difficult for me to judge the method used to generate these RNA velocity maps.*

Thank you! The RNA velocity analysis conforms to the state of the art.

*6) Line 191 “A large number of genes indeed varied between *lakritz* and WT” I could not find the figure associated with this statement.*

We had removed a multi-panel supplemental figure in the current version to avoid overloading the manuscript with data that are not followed up with experiments. To address the reviewer’s suggestion, we added a new figure (Fig. 7) showing representative HCR data for three informative markers (*fabp7a*, *shha* and *ptch1*), for which we see biological relevance.

7) Line 195, “While, in some cases, PCs are ranked differently, the top PCs are identical across groups (Supplementary Figs. 11a,b)” given the fact that these top PCs explain only the big variances, is this a good method really to use in order to say there are no changes in expression patterns ? I don’t have strong opinion about this figure, but I am not sure how strong this figure adds to the support of the major claim.

We agree that the top PCs explain the large sources of variance between groups. This is especially true for the 1st and 2nd PCs. Considered alone, this would not support our claim that there is little difference in cell types between the two groups. However, we do believe that some readers will find our results easier to understand by increasing levels of granularity.

Reviewer #2:

The manuscript by Sherman and colleagues is an interesting study investigating the question how a sensory part of the brain develops in the absence of sensual input. The picked an

interesting system, namely the zebrafish lakritz mutant, which does not develop retinal ganglion cells and hence retinorecipient areas of the brain develop devoid of external input. The authors used a sophisticated and exhaustive transcriptional profiling. Surprisingly they found very little transcriptional changes among the 77 neuronal types they identified. A few individual genes were slightly misregulated, but the neuronal types developed in normal proportions and locations. Somewhat akin to the prolongation of critical periods during sensual deprivation, cell cycles were slower and terminal differentiation in the brain is slowed down.

Although the result is not completely surprising, given the apparent normal development of activity blocked zebrafish embryos in classical experiments, the extend of normal neural circuit assembly is astonishing. A nice icing of the cake is the final experiment, where the authors optogenetically stimulate pretectal neurons to evoke proper eye movement in the absence of visual input.

I only have few minor comments that the authors may consider. It is not clear to the reviewer why in the J-turn experiment morphants instead of mutants (or crispants) have been used.

We decided to use an *atoh7* morpholino in this experiment instead of the mutation for efficiency reasons: It would have taken close to a year to generate homozygous mutants with the appropriate triple-transgenic background. The specific morpholino sequence we chose has been shown in multiple studies to eliminate the generation of RGCs up until day 5 by Chi-Bin Chien's group (Pittman et al. 2008). To make this choice more transparent, we added the following explanation to the methods section: "For this experiment, *atoh7* morphants were used instead of *lakritz* mutants, as they have been shown to exactly phenocopy the mutant at embryonic and larval stages (Pittman et al. 2008, Gupta et al. 2018)."

In Figure 7e, a dot plot would have been more informative than a box plot.

We believe that a box plot is a straightforward way to convey that *lakritz* mutants are able to perform an OKR. Please note that we cannot (and do not!) claim that the behavior is perfectly developed in *lakritz* mutants. A quantitative comparison is beyond the scope of the present study and of limited interest, since we cannot perfectly reconstitute the sensory-evoked activity pattern by optogenetics.

Overall this is a very interesting paper with extensive and sensible use of data analysis of single transcriptome data, beautiful anatomical information, and a very nice behavioral confirmation that neural circuit formation is independent of sensory input.

Thank you.

Reviewer #3:

The manuscript of Sherman et al. demonstrates that inputs and signals from zebrafish retina are not necessary for the establishment of neuron subtypes (molecular identities) and functional synaptic connections in the brain areas which are involved in visual processing and responses. This is a significant and original contribution to the fundamental question of to what extent brain

maturation is shaped by external stimuli.

The model organism is a zebrafish lakritz mutant (atoh7^{-/-}) that lacks retinal connections, established in a laboratory of the senior author over 20 years ago and well characterised. Single cell RNA-seq followed by a complex bioinformatics analysis convincingly show that developmental trajectories and diversity of neurons were not affected. This agrees with recent data on mice, and extends the conclusion from this previous study, because excludes the potential role of factors secreted from retinal connections. Furthermore, optogenetic stimulation of pretectal areas evoked normal visual behaviours, credibly demonstrating that central visual circuits developed normally and were functional. This result is unexpected and exciting because previous research, mainly on mammals, showed that a disruption of visual input alters the formation of visual maps and wiring of the visual system. Overall, I find this study very interesting, especially for neurodevelopment researchers. Additionally, transcriptomic data with spatial resolution, which are a part of this study, provide very useful information about markers which can be used to distinguish neuronal sub-populations in zebrafish, and possibly also in other vertebrates.

The manuscript is generally well written. The title is adequate, the Abstract is a good summary of the study, the Introduction is concise and informative, interpretations and conclusions in the Discussion are clear and justified. My concerns (major flaws) are directed mainly to the structure of the Result section.

Major flaws

1) In Results, I see a problem with an overload with bioinformatics data. Three chapters (lines 140-225) correspond to figures 3 and 4. While these two figures are barely mentioned, other twelve (supplementary) figures are quite detailedly described, discouraging from further reading. Actually, one of these chapter cites only supplementary figures. If these results were not crucial enough to be in the main text why to devote so large amount of space to them? I understand that all these impressive bioinformatics analyses were very complex and necessary to draw strong conclusions, but the majority of readers probably would not follow them. I wonder if it would be possible to move some of these descriptions to Methods or to the supplement, not to overshadow the message with results which are supportive.

I also think that Fig. 3 a-b could be moved to supplements, Fig3c-d could be merged with Fig. 4, and these results could be compactly described in one chapter with one bigger message. Similar situation is with Fig. 5 and 6. Are the panels 6a, b, c, d necessary for the main text? Could be the two chapters corresponding to figures 5 and 6 merged into one concise chapter?

We appreciate the reviewer's suggestions for improvement of the data presentation. We have streamlined the text and moved details to the Methods, following the reviewer's advice. However, we prefer to keep Figures 3 to 6 as they are. The target audience for our paper will be both bioinformatics experts and neurobiologists, of whom the latter may not be as steeped in single-cell transcriptomic analyses as the reviewer seems to be. Figures 3 to 6 illustrate the depth of our analysis and should therefore be prominently visible. Moving individual panels to the supplemental data, which, as the reviewer points out, are already daunting in their density and complexity, would not make the story more accessible.

2) The chapter “Expression of progenitor and early precursor...” cites only supplementary tables. Moreover, it describes a panel of *in situ* hybridisations, but does not show any of these stainings. They are only summarised in the tables in the form of numbers from statistical analysis. Representative images should be provided. Also, the quantitative analysis, though well done from the methodological point of view, does not convince me, because the differences are usually small, up to 1.5-fold change. I am sceptical if *in situ* hybridisation can show such differences reliably. In my opinion these results raise doubts rather than reinforce conclusions from RNA-seq. I would rather not place them in this paper.

We have condensed and streamlined this section and now include a representative image showing reduction of *fabp7a* in *lakritz* mutants, following the reviewer’s advice. For the progenitor marker *fabp7a*, we found the most consistent and strongest dysregulation across retinorecipient regions. This finding is presented in a new figure (Fig. 7) showing representative HCR data for three biologically relevant markers (*fabp7a*, *shha* and *ptch1*).

The HCR method is widely accepted as a quantitative (or at least semi-quantitative) readout of transcript level. However, because the reviewer’s general skepticism is valid, we consider differences in gene expression level merely as trends. Still, we wish to show the results as a supplemental table, if only to signal to the community that these HCR probes exist and are available upon request.

3) The conclusions about cell cycle exit delay are based solely on single cell RNA-seq analysis. I suggest using another method to confirm this claim. Is it possible to analyse cell cycle by a *puls BrdU* incorporation combined with *in situ* hybridisation or FACS?

While these conclusions are based on single-cell transcriptomics, we employed different methods that converge on the same result. Additionally, the quantitative HCR analysis picks up the largest effects in the dysregulation of *fabp7a*, as was predicted by our RNA velocity and cell-cycle analysis. As multiple analyses agree on this finding, further experiments are unlikely to add new insights.

Minor flaws

1) Line 80-81: “Supplementary Figs. 1b-f “– I suppose it was to be „Figs 1b-f”

Corrected.

2) Fig. 1 legend – to make this figure+legend self-explanatory, I suggest including the information that green cells were sorted for subsequent single-cell RNA-seq

Changed figure legend.

3) Fig. 1 g-h – would it be possible to mark which cell cluster belongs to which brain part? for example in the x-axis.

It is difficult to associate all clusters with specific brain areas. To fully resolve this computationally we would need to perform spatial transcriptomics and recluster the data with this added dimension. Table S1, however, partially addresses this by summarizing the top anatomical hits for the expression of each gene.

4) Fig. 2 looks a bit messy. Do you need figure 2b, which is a repetition of fig. 1a? Why GABAergic clusters are not shown for 2a and 2f? Why there are no gene labels in the right panel, unlike the left and medial panels?

Each row in this figure matches the expression of two genes. This figure shows the power of combining both single-cell transcriptomic profiling with HCRs. Fig. 1a shows the transgene expression pattern in a single fish in the context of the head and brain. Fig. 2b shows the averaged transgene pattern from the mapzebrain atlas. We felt that readers will benefit from having the anatomical annotations accessible in panel 2b. We did not show all GABAergic UMAPs, because the markers in panels 2a and 2f are only expressed in glutamatergic neurons.

5) PTU abbreviation should be explained.

Added clarification for abbreviation.

6) Information that “all probes were purchased from the manufacturer” should be supplemented with at least the name of the manufacturer.

Added clarification.

Loose suggestion

1) While the first chapters of Introduction are excellent, the last Introduction chapter, which practically describes the results, is weaker. I find it too long, and at the same time it lacks a clear question. I suppose it could be easily improved.

We have revised and shortened the Introduction accordingly.

2) The Discussion section lacks a closure. The last sentence serves this purpose, but one paragraph with a couple of sentences would do the job better

We have reorganized the Discussion to flesh out the concluding paragraph without repeating any of the previous points. We think it reads much better now – thank you for the suggestion!

REVIEWERS' COMMENTS

Reviewer #1 (Remarks to the Author):

I read the revised manuscript and the responses of the authors to my comments, as well as other reviewers comments.

The authors successfully addressed all my comments.

I support the publication of manuscript in Nature Communications

Reviewer #2 (Remarks to the Author):

I am happy with the implemented changes. This is an interesting paper that will be well received by the community.

Reviewer #3 (Remarks to the Author):

In the revised version of the manuscript, the Authors addressed all my comments and introduced satisfactory corrections. They did not fully agree with me on some points, but their explanations convinced me completely. I stand by my previous opinion that the manuscript presents exciting research and has a clear message, and now it reads even better. I recommend this paper for publication in Nature Communication.